# The Meta-Evaluation Problem in Explainable AI: Identifying Reliable Estimators with `MetaQuantus`

**Anna Hedström**[1,6,†]             *anna.hedstroem@tu-berlin.de*
**Philine Bommer**[1,6]             *philine.bommer@tu-berlin.de*
**Kristoffer K. Wickstrøm**[3]          *kristoffer.k.wickstrom@uit.no*
**Wojciech Samek**[1,2,4]         *wojciech.samek@hhi.fraunhofer.de*
**Sebastian Lapuschkin**[4]       *sebastian.lapuschkin@hhi.fraunhofer.de*
**Marina M.-C. Höhne**[2,3,5,6,†]         *mhoehne@atb-potsdam.de*

[1] *Department of Electrical Engineering and Computer Science, TU Berlin*
[2] *BIFOLD – Berlin Institute for the Foundations of Learning and Data*
[3] *Department of Physics and Technology, UiT the Arctic University of Norway*
[4] *Department of Artificial Intelligence, Fraunhofer Heinrich-Hertz-Institute*
[5] *Department of Computer Science, University of Potsdam*
[6] *UMI Lab, Leibniz Institute of Agricultural Engineering and Bioeconomy e.V. (ATB)*
[†] *corresponding authors*

**Reviewed on OpenReview:** <https://openreview.net/forum?id=j3FKOOHyfU>

## Abstract

One of the unsolved challenges in the field of Explainable AI (XAI) is determining how to most reliably estimate the quality of an explanation method in the absence of ground truth explanation labels. Resolving this issue is of utmost importance as the evaluation outcomes generated by competing evaluation methods (or "quality estimators"), which aim at measuring the same property of an explanation method, frequently present conflicting rankings. Such disagreements can be challenging for practitioners to interpret, thereby complicating their ability to select the best-performing explanation method. We address this problem through a meta-evaluation of different quality estimators in XAI, which we define as *"the process of evaluating the evaluation method"*. Our novel framework, `MetaQuantus`, analyses two complementary performance characteristics of a quality estimator: its resilience to noise and reactivity to randomness, thus circumventing the need for ground truth labels. We demonstrate the effectiveness of our framework through a series of experiments, targeting various open questions in XAI such as the selection and hyperparameter optimisation of quality estimators. Our work is released under an open-source license[1] to serve as a development tool for XAI- and Machine Learning (ML) practitioners to verify and benchmark newly constructed quality estimators in a given explainability context. With this work, we provide the community with clear and theoretically-grounded guidance for identifying reliable evaluation methods, thus facilitating reproducibility in the field.

## 1 Introduction

Since Explainable AI (XAI) is intended to increase trust and transparency in AI systems, it is necessary to evaluate the performance of proposed explanation methods to ensure their reliability. In the context of black-box Machine Learning (ML) models such as neural networks (NNs)—where the input to output mapping is not explicitly known or interpretable by a user (Benitez et al., 1997; Bellido & Fiesler, 1993; Lipton, 2018)—there is generally an absence of ground truth explanations to understand the model's decision. This makes it difficult to evaluate the performance of explanation methods since the exact outcomes of explanations

---

[1]Code is available at the GitHub repository: <https://github.com/annahedstroem/MetaQuantus>.

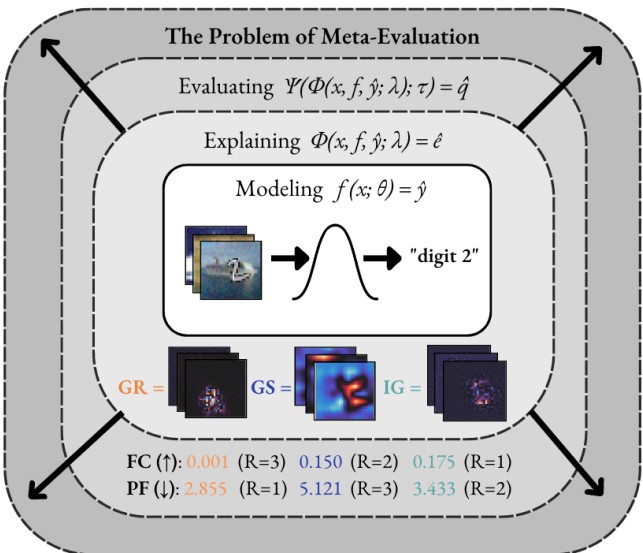

Figure 1: An illustration of Meta-Evaluation through three phases: (i) Modeling, (ii) Explaining and (iii) Evaluating. (i) A ResNet-9 model (He et al., 2016) is trained to classify digits from 0 to 9 on Customised-MNIST dataset (Bykov et al., 2022) (i.e., MNIST digits pasted on randomly sampled CIFAR-10 backgrounds). (ii) To understand the model's prediction, we use several explanation methods including *Gradient* (Morch et al., 1995; Baehrens et al., 2010), *Integrated Gradients* (Sundararajan et al., 2017) and *GradientShap* (Lundberg & Lee, 2017), which are distinguished by their respective colours. (iii) To evaluate the quality of the explanations, we apply different estimators of faithfulness such as *Faithfulness Correlation* (FC) (Bhatt et al., 2020) and *Pixel-Flipping* (PF) (Bach et al., 2015), which return a correlation coefficient and an AUC score, respectively. However, since the scores vary depending on the estimator, both in range and direction, with lower or higher scores indicating more faithful explanations, interpreting the resulting faithfulness scores remains difficult for the practitioner.

oftentimes remain unknown and thus unverifiable (Dasgupta et al., 2022). Without consensus around how to define the quality or "correctness" of an explanation method, a variety of XAI evaluation procedures have been proposed. These efforts most commonly involve (i) measuring the extent to which desirable properties are fulfilled by the explanation method, e.g., through faithfulness or robustness analysis (Samek et al., 2017; Sundararajan et al., 2017; Montavon et al., 2018; Agarwal et al., 2022a), (ii) generating well-defined, synthetic settings where explanation labels are simulated (Yang & Kim, 2019; Arras et al., 2022; Liu et al., 2021a) or, (iii) evaluating explanations based on visual alignment with a human prior (Smilkov et al., 2017). Most relevant to our work is the first category of evaluation methods or "metrics" (Hedström et al., 2023), where the goal is to estimate the quality of an attribution-based explanation method, also known as feature-importance method. Henceforth, we refer to these evaluation methods as "quality estimators", or simply "estimators" of explanation quality.

The abundance of explanation methods and an ever-growing number of quality estimators, combined with little guidance on how to use them, have caused practitioner confusion within the XAI and ML communities Krishna et al. (2022). Strong claims of identified failure modes for explanation methods with assertions of which methods pass or fail (Adebayo et al., 2018; Dombrowski et al., 2019; Sixt et al., 2020), followed by rebuttals (Sundararajan & Taly, 2018; Yona & Greenfeld, 2021; Binder et al., 2022), are ever-present. To answer the question of "which explanation method to use for a given task", we must first be able to answer "how to reliably define and measure the relevant qualities that an explanation method should fulfill". While preliminary efforts exist to address this issue (Brunke et al., 2020; Tomsett et al., 2020; Gevaert et al., 2022; Rong et al., 2022), to the best of our knowledge, there is currently no comprehensive solution that thoroughly evaluates the various estimators used to compare, select and reject different explanation methods in XAI. Previous efforts at addressing this issue have been limited in scope and do not provide a thorough theoretical motivation. With this work, we aim to fill this critical yet largely neglected research gap.

In this work, we characterise the problem of meta-evaluation, which we refer to as *"the process of evaluating the evaluation method"*. This problem arises as we select and quantitatively evaluate different explanation methods for a given model, dataset, and task—and where conflicting evaluation outcomes are produced. As illustrated in Figure 1, we can apply various estimators to compare the explanation methods' faithfulness, which measures how closely the explanations align with the predictive behaviour of the model (the experimental details are described in Appendix A.4). However, the estimators rank the same explanation methods differently, e.g., the *Gradient* method (Morch et al., 1995; Baehrens et al., 2010) is both ranked the highest (R=1) and the lowest (R=3) depending on the estimator used. With a disagreement in evaluation outcome about which explanation method is superior (Krishna et al., 2022) coupled with little to no guidance on how to identify a high-quality estimator (Wang & Wang, 2022), practitioners may unknowingly choose an inferior estimator which ultimately results in a selection of an explanation method that presents a less faithful explanation to the end user.

To address the issue of evaluation disagreement in XAI, we propose a simple yet comprehensive framework which primary purpose is to provide an objective, independent view of the estimator's performance by meta-evaluating it against two failure modes: resilience to noise (NR) and reactivity to adversary (AR). Similar to how software systems undergo vulnerability- and penetration tests before getting deployed in a larger system, we apply this framework to stress test the estimators. If vulnerabilities in the quality estimator are discovered, e.g., high sensitivity to noise in input or low reactivity to randomness, appropriate actions can be taken to improve the estimators. The contribution of this work is three-fold.

- First, we provide a clear argument for why meta-evaluation of quality estimators is challenging (Section 2.2), emphasising the importance of reliability analysis as a tool to analyse estimator behaviour.

- Second, based on these findings, we propose a framework to meta-evaluate the performance of XAI estimators (Section 3), with a sound theoretical foundation and agnosticism towards various network architectures and attribution-based explanation methods.

- Third, through experiments on various explanation methods, datasets, and models, we demonstrate that our framework can solve a range of XAI-related tasks such as estimator selection and hyperparameter optimisation, generating novel insights into estimators' behaviour (Section 6).

We find it important to point out that we have no interest in developing yet another quality estimator. The real need in our community lies in developing meta-evaluation schemes to validate and determine the reliability of the quality estimators that already been developed. It is surprising to us that very little effort has so far been directed towards this important area of analysing their performance. With this work, we aim to support the process of selecting a quality estimator in a given explainability context in order to provide more clarity and guidance on how to effectively evaluate explanation methods.

## 1.1 Related Works

Despite much activity towards the development of estimators to evaluate explanation quality (Bach et al., 2015; Sundararajan et al., 2017; Bhatt et al., 2020; Nguyen & Martinez, 2020; Rieger & Hansen, 2020; Arias-Duart et al., 2021) and benchmarking tools (Liu et al., 2021a; Agarwal et al., 2022b; Hedström et al., 2023), limited attention has thus far been given to analysing the performance of the estimators themselves. Only recently, increased attention has been raised on the intricacies that come with XAI evaluation, for example, the contributions of Brunke et al. (2020); Brocki & Chung (2022); Rong et al. (2022) emphasise the difficulty that comes with parameterising estimators. Another issue with evaluation was brought to light by Neely et al. (2021); Krishna et al. (2022), which revealed that explanations frequently disagree in their ranking of features. Additionally, several independent research groups were able to identify empirical "confounders" (Sundararajan & Taly, 2018; Kokhlikyan et al., 2021; Yona & Greenfeld, 2021; Binder et al., 2022) affecting the well-adopted *Model Parameter Randomisation* test (Adebayo et al., 2018). From these publications, it seems worryingly "easy to get it wrong" when it comes to evaluating explainable methods empirically. There still remains a lot of ambiguity when it comes to determining what makes up a good or bad quality estimator (Wang & Wang, 2022).

Within the scope of evaluating quality estimators, preliminary efforts exist but a unified effort is required. Since there is little to no consensus on how to determine the true value of a quality estimator—when a new estimator is introduced, it is often assessed based on single perspectives, e.g., by randomisation experiments (Rieger & Hansen, 2020; Arias-Duart et al., 2021) or ranking consistency (Rong et al., 2022). All of these mentioned works are undoubtedly steps in the right direction, but what is missing is a broader, more comprehensive framing of what a quality estimator ought to fulfil. With this work, we aim to fill this gap.

## 2    Preliminaries

In the following, we derive a mathematical definition of the evaluation problem in XAI by outlining the key elements required to perform quality estimation on a given explanation method. In the succeeding section, we discuss the Challenge of Unverifiability (CoU) which explains why meta-evaluation is theoretically difficult. All notation used throughout this paper can be found in Appendix A.7.

### 2.1    The Evaluation Problem

Consider a supervised classification problem[2] where we have a black-box model $f$ parameterised by $\theta$ that has been trained on a given training dataset $\boldsymbol{X}_{\mathrm{tr}} = \{(\boldsymbol{x}_1, y_1), \ldots, (\boldsymbol{x}_N, y_N)\}$ to map an input $\boldsymbol{x} \in \mathbb{R}^D$ to an output class $y \in \{1, \ldots, C\}$, with a trained functional mapping such as:

$$f(\boldsymbol{x}; \theta) = \hat{y}, \tag{1}$$

More generally, we can define the model function $f : \mathbb{X} \mapsto \mathbb{Y}$ that maps inputs from the instance space $\mathbb{X}$ to predictions in the label space $\mathbb{Y}$ with $\boldsymbol{x} \in \mathbb{X}$ and $\hat{y} \in \mathbb{Y}$. Let $\mathbb{F}$ denote the function space such that $f \in \mathbb{F}$. To quantitatively estimate the performance of model $f$, we compute the prediction error on a given test dataset $\boldsymbol{X}_{\mathrm{te}}$ where there exists a label $y$ for each prediction $\hat{y}$. To understand the reasoning of the model $f$ behind a certain prediction $\hat{y}$, we can apply one of the many proposed *local* explanation methods (Smilkov et al., 2017; Zeiler & Fergus, 2014; Sundararajan et al., 2017; Selvaraju et al., 2020; Bykov et al., 2022) as follows:

$$\Phi(\boldsymbol{x}, f, \hat{y}; \lambda) = \hat{\boldsymbol{e}}, \tag{2}$$

where $\Phi : \mathbb{R}^D \times \mathbb{F} \times \mathbb{Y} \mapsto \mathbb{R}^D$ is an explanation function that is parameterised by $\lambda$ and which distributes an attribution to each individual feature in $\boldsymbol{x}$ according to its importance, typically visualised in an explanation map $\hat{\boldsymbol{e}} \in \mathbb{R}^D$. A broad variety of explanation methods fall within the scope of $\Phi$ such as gradient-based (Smilkov et al., 2017; Sundararajan et al., 2017; Bykov et al., 2022), back-propagation-based (Bach et al., 2015), model-agnostic (Zeiler & Fergus, 2014; Lundberg & Lee, 2017), local surrogate (Ribeiro et al., 2016), attention-based (Chefer et al., 2021; Covert et al., 2022), as well as prototypical explanation methods (Simonyan et al., 2014). Let $\mathbb{E}$ denote the space of possible explanations with respect to the model such that $\Phi \in \mathbb{E}$.

Similar to how we compute the prediction error to estimate the performance of a model $f$, to evaluate the quality of the explanation function $\Phi$, we compute the explanation error, requiring a ground truth explanation $\boldsymbol{e}$. These labels are, however, generally not available for black-box ML models and in particular NNs[3] since their inner workings are considered uninterpretable (Bellido & Fiesler, 1993; Benitez et al., 1997; Dasgupta et al., 2022). Therefore, XAI researchers and ML practitioners must resort to indirect approaches to estimate the quality of a given explanation, e.g., by measuring the explanation's relative fulfilment of certain human-defined properties. Recent work by Hedström et al. (2023) has proposed to group these properties of explanation quality into six categories; (a) faithfulness, (b) robustness, (c) localisation, (d) randomisation, (e) complexity, and (f) axiomatic estimators which provide a natural framework to compare and analyse explanation quality. A summary of these explanation quality categories can be found in Appendix A.2 (see Equations 11-14).

---

[2]Since classification tasks are commonly encountered in the XAI community, it is chosen to illustrate the Evaluation Problem. However, as discussed in Appendix A.1.1, our statements also apply to other prediction scenarios.

[3]Even in toy- or expert-annotated ground truth XAI datasets where features are known a priori (Arras et al., 2022; Yang & Kim, 2019), the explanation labels provided may not accurately reflect the model's decision-making process. Only with perfect knowledge of the model can true ground truth explanation labels be obtained.

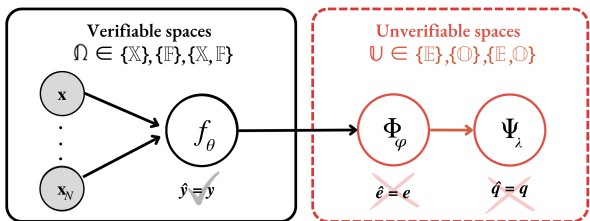

Figure 2: A visual representation of the conditional dependencies between variables in quality estimation (Equation 3). The information flows from modelling to explaining and evaluating the explanations, i.e., $\Psi \circ \Phi \circ f$, which is indicated by the direction of the arrows in the directed acyclic graph (DAG). The colours indicate if the spaces have verifiable (black) or unverifiable outcomes (red).

We provide a generalised notation for quality estimation of attribution-based explanation methods as follows. Let $\Psi_\tau : \mathbb{E} \times \mathbb{R}^D \times \mathbb{F} \times \mathbb{Y} \mapsto \mathbb{R}$ be a quality estimator that is parameterised by $\tau$ and takes one explanation and returns one scalar value ("quality estimate") to indicate the quality of the explanation. The evaluation of an explanation, i.e., quality estimation, can be written as follows:

$$\Psi(\Phi, \boldsymbol{x}, f, \hat{y}; \tau) = \hat{q}, \tag{3}$$

where $\Psi$ represents the quality estimator and the whole space of possible estimators is denoted $\Psi \in \mathbb{O}$. Mathematical descriptions of such estimators can be found in Appendix (see Equations A.3).

## 2.2 The Challenge of Unverifiability

The goal of quantitative XAI evaluation is to provide an objective measure of the quality of an explanation. However, due to missing ground truth, the quantitative assessment of neural network explanations remains non-trivial. To clarify where this difficulty arises, we represent the process of quality estimation as a directed acyclic graph (DAG), as seen in Figure 2. Here, each node represents a random variable and the edges represent the relationships between the variables, with the uncertainty of a parent node propagating to its child node. We separate the nodes between verifiable- and unverifiable spaces. The verifiable spaces are spaces where ground truth labels are available, i.e., $\mathbb{\cap} \in \{\{\mathbb{X}\}, \{\mathbb{F}\}, \{\mathbb{X}, \mathbb{F}\}\}$ and the unverifiable spaces include spaces where there is an absence of labels, i.e., $\mathbb{U} \in \{\{\mathbb{E}\}, \{\mathbb{O}\}, \{\mathbb{E}, \mathbb{O}\}\}$.

As indicated by the direction of the arrows, a key observation is that in quality estimation there exists a conditional dependency between the variables of modelling, explaining and evaluating (the explanations). This further means that since the evaluation function is applied to the results of the unverifiable explanation function, the evaluation outcome also renders unverifiable. We refer to this phenomenon as the Challenge of Unverifiability. Another key observation is that we cannot determine the accuracy or validity of an estimator (i.e., whether it actually measures the intended quality) since such an assessment requires access to ground truth labels. However, as reliability analysis does not depend on the availability of ground truth labels, it is still possible to study the reliability of an estimator, which refers to its overall consistency ("does this estimator produce similar results under consistent conditions?"). This can be achieved by repeatedly measuring the evaluation outcomes that result from fixing the unverifiable parameters and functions and only varying the elements of the verifiable spaces. In the following, we will use the distinction between verifiable- and unverifiable spaces to systematically and controllably measure the performance of quality estimators.

## 3 A Meta-Evaluation Framework

While the Challenge of Unverifiability makes meta-evaluation of quality estimators challenging, it is still possible to study the performance characteristics of an estimator through the lens of reliability. To this end, we developed a three-step framework, which is a higher-level evaluation scheme that examines quality estimators that themselves have been used to evaluate a particular explanation method.

### 3.1 Defining Failure Modes

Without ground truth information, we cannot validate or optimise the quality estimators against what we want them to fulfil, but we can instead articulate edge-case scenarios or behaviours that we do not want them to exhibit. For this purpose, we formulate failure modes which are described in the following.

**Failure Mode 1** (Noise Resilience)**.** *A quality estimator should be resilient to minor perturbations of its input parameters.*

Similar to the stability (or "robustness") property of explanation functions (Agarwal et al., 2022a; Montavon et al., 2018) and especially *Lipschitz Continuity* (Alvarez-Melis & Jaakkola, 2018; Yeh et al., 2019), where small changes in the input should only lead to small changes in the explanation, noise resilience (NR) evaluates the extent to which a quality estimator is robust towards minor perturbations of its inputs. Following our general perturbation Definition 3 in Appendix A.2, we define a minor perturbation $\mathcal{P}_{\Cap}^{M}$ of any verifiable space $\Cap$ as follows:

**Definition 1** (Minor Perturbation)**.** Let $\mathcal{P}_{\Cap}(\boldsymbol{\omega})$ be a perturbation function of $\boldsymbol{\omega} \in \Cap$, $\hat{y} = f(\boldsymbol{x}; \theta)$ be the original prediction of the network and $y'$ be the prediction after the perturbation. Then $\mathcal{P}_{\Cap}(\boldsymbol{\omega})$ is minor $\mathcal{P}_{\Cap}^{M}$, if $\forall \, \hat{y}, \, y' \in \{\{f(\mathcal{P}_{\mathbb{X}}^{M}(\boldsymbol{x}); \theta)\}, \{f(\boldsymbol{x}; \mathcal{P}_{\mathbb{F}}^{M}(\theta))\}, \{f(\mathcal{P}_{\mathbb{X}}^{M}(\boldsymbol{x}); \mathcal{P}_{\mathbb{F}}^{M}(\theta))\}\}$, $\exists \, \epsilon \in \mathbb{R} \, \epsilon \ll 1$ such that:

$$||\hat{y} - y'||_p \leq \epsilon$$

For classification, we employ L1-norm with $p = 1$, thus, Definition 1 states that the predicted label $y'$ stays unchanged after the perturbation, i.e., $\hat{y} \approx y'$. Similar to works by Brunke et al. (2020); Brocki & Chung (2022); Rong et al. (2022), we measure the vulnerability of quality estimators to variations or "minor confounds" in the estimator. However, in contrast to these aforementioned works, we only perturb in the verifiable space by means of measuring the change in the model decision on a sample before and after perturbation, and thus we can control and directly measure the strength of the perturbation. Accordingly, to quantitatively examine Failure Mode 1, we expose the estimator to perturbations with small or minor impacts. Complementary to testing an estimator's resilience to noise, we also formulate a second failure mode to test whether a quality estimator produces a significant change when exposed to disruptive perturbation, i.e., randomisation to any of its inputs.

**Failure Mode 2** (Adversary Reactivity)**.** *A quality estimator should be reactive to disruptive perturbations of its input parameters.*

Previous research has noted that the estimators' scores should be conceivably different when produced for a random explanation (Rieger & Hansen, 2020) or a randomly initialised model (Arias-Duart et al., 2021). Our approach is similar in that it also seeks to disrupt the explanation process. However, since we can control the perturbation strength in the verifiable spaces, we can make more well-grounded claims about the expected outcomes of a perturbation. Theoretically, we define disruptive perturbations $\mathcal{P}_{\Cap}^{D}$ contrary to Definition 1.

**Definition 2** (Disruptive Perturbation)**.** $\mathcal{P}_{\Cap}(\boldsymbol{\omega})$ be a perturbation function of $\boldsymbol{\omega} \in \Cap$, $\hat{y} = f(\boldsymbol{x}; \theta)$ be the original prediction of the network and $y'$ be the prediction after the perturbation. Then $\mathcal{P}_{\Cap}(\omega)$ is disruptive $\mathcal{P}_{\Cap}^{D}$, if $\forall \, \hat{y}, \, y' \in \{\{f(\mathcal{P}_{\mathbb{X}}^{D}(\boldsymbol{x}); \theta)\}, \{f(\boldsymbol{x}; \mathcal{P}_{\mathbb{F}}^{D}(\theta))\}, \{f(\mathcal{P}_{\mathbb{X}}^{D}(\boldsymbol{x}); \mathcal{P}_{\mathbb{F}}^{D}(\theta))\}\} \exists \, \epsilon \in \mathbb{R}, \, \epsilon \ll 1$ such that:

$$||\hat{y} - y'||_p > \epsilon.$$

In a classification context, Definition 2 implies a change in the predicted class label. Figure 3 illustrates the main difference between minor and disruptive perturbations, which is that the decision boundary remains uncrossed or crossed, respectively. In Appendix A.1.1, we expand the Definitions 1 and 2 to other problem settings such as multi-label classification and also discuss how adversarial attacks relate to these definitions.

Using Definitions 1 or 2, we can generate perturbed quality estimates $q'$ by applying a minor or disruptive perturbation on the verifiable spaces in the input, model, or input- and model spaces simultaneously:

$$\hat{q} \in \{ \; \Psi(\Phi, \mathcal{P}_{\mathbb{X}}^{t}(\boldsymbol{x}), f, \hat{y}), \;\; \Psi(\Phi, \boldsymbol{x}, \mathcal{P}_{\mathbb{F}}^{t}(\theta), \hat{y})), \;\; \Psi(\Phi, \mathcal{P}_{\mathbb{X}}^{t}(\boldsymbol{x}), \mathcal{P}_{\mathbb{F}}^{t}(\theta), \hat{y})) \; \}, \tag{4}$$

where the superscript of the perturbation function, $t \in \{M, D\}$ indicates the perturbation strength. For simplicity, we omit the hyperparameters $\tau, \lambda$ from Equation 4. By repeating this perturbation (Equation 4)

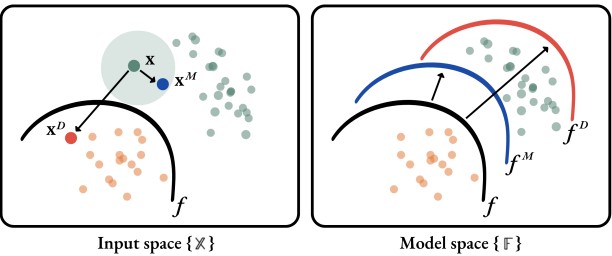

Figure 3: An illustration of minor versus disruptive perturbation in the different spaces (left: $\mathbb{X}$, right: $\mathbb{F}$) for a classification task. The direction of the arrows shows how the respective perturbations are realised, where blue and red colours indicate a minor- or disruptive perturbation, respectively. The minor perturbation keeps the decision boundary intact, either by perturbing a sample $\boldsymbol{x}^M$ (left) or perturbing the model itself $f^D$ (right). The disruptive perturbation implies that the decision boundary is crossed either through a sample $\boldsymbol{x}^D$ (left) or model $f^D$ (right).

multiple times, we gather sets of perturbed estimates for meta-evaluation analysis. In the next section, we provide a detailed description of how this analysis is performed.

## 3.2 Formulating Consistency Criteria

To determine whether a quality estimator appropriately circumvented a failure mode, we can measure the similarity of its quality estimates before and after the perturbation. After a minor perturbation, when testing for noise resilience, we would expect that the scores are similarly distributed. Conversely, for disruptive perturbations, when testing for reactivity to adversary, we would anticipate a large response to information annihilation of the explanation process by means of scores being dis-similarly distributed. We formalise this idea in our **I**ntr**a**-**C**onsistency (IAC) criterion as follows:

$$\mathbf{IAC} = \frac{1}{K} \sum_{k=1}^{K} d(\hat{\boldsymbol{q}}, \boldsymbol{q}_k'), \tag{5}$$

where $\hat{\boldsymbol{q}}$ refers to unperturbed estimates and $\boldsymbol{q}_k' \in \mathbb{R}^N, k = (1, \ldots, K)$ is a set of perturbed quality estimates, replicated $K$ times for $N$ test samples (see Equation 4) such that $\boldsymbol{Q} = [\boldsymbol{q}_1', \ldots, \boldsymbol{q}_K'] \in \mathbb{R}^{N \times K}$. Here, $d$ refers to a statistical significance measure $d : \mathbb{R}^N \times \mathbb{R}^N \mapsto \mathbb{R}$ that takes a set of unperturbed- and perturbed estimates and returns a p-value. A high p-value indicates that $\hat{\boldsymbol{q}}$ and $\boldsymbol{q}_k'$ are similarly distributed and a low p-value value means that the estimates are differently distributed. Accordingly, Equation 5 returns the average p-value across all perturbed samples over $K$ perturbations, with IAC $\in [0, 1]$. Since the nominal values of quality estimators can vary and often have little to no semantic meaning, we use the non-parametric *Wilcoxon signed-rank test* (Wilcoxon, 1945) which does not carry strong assumptions about the data distribution, only about its ranking. In addition to the intra-consistency analysis, we also measure whether quality estimators exhibit consistent behaviour in terms of ranking. This type of inter-consistency analysis is commonly used in Explainable AI research (Tomsett et al., 2020; Gevaert et al., 2022; Hedström et al., 2023; Rong et al., 2022) and complements the aforementioned by involving more than one explanation method. Let $\bar{\boldsymbol{Q}} \in \mathbb{R}^{N \times L}$ denote a matrix for the unperturbed estimates $\hat{\boldsymbol{q}}$ for $L$ explanation methods and $\bar{\boldsymbol{Q}}' \in \mathbb{R}^{N \times L}$ be a matrix for the perturbed estimates $\boldsymbol{q}_k'$, which are both averaged over $K$ perturbations. We formulate the **I**nt**e**r-**C**onsistency (IEC) criterion as follows:

$$\mathbf{IEC} = \frac{1}{N \times L} \sum_{i=1}^{N} \sum_{j=1}^{L} U_{i,j}^t \tag{6}$$

where $U_{i,j}^t \in [0, 1]$ are entries of a binary ranking agreement matrix $\boldsymbol{U}$ that takes quality estimates from $\bar{\boldsymbol{Q}}$ and $\bar{\boldsymbol{Q}}'$ and populates the entries according to the interpretation of ranking. Here, IEC = 1 indicates perfect ranking consistency and IEC = 0 the absence of it, where IEC $\in [0, 1]$. The perturbation strength is indicated in the superscript $t \in \{M, D\}$. The interpretation of ranking is different depending on the perturbation strength, i.e., minor or disruptive. For minor perturbations, we measure if the quality estimator

**Step 1. Perturbing**

*Depending on failure mode, initiate a minor or disruptive perturbation*

$\mathcal{P}_\Omega^M$ — Noise Resilience →

$\mathcal{P}_\Omega^D$ — Adversary Reactivity →

**Step 2. Scoring**

*Measure effects of the perturbations via IAC and IEC criteria*

$$\text{IAC} = \frac{1}{K}\sum_{k=1}^{K} d(\hat{q}, q_k'), \quad (5)$$

$\text{IAC}_{NR}$  — Perturbed scores / Original scores

$\text{IAC}_{AR}$  — Perturbed scores / Original scores

$L = $ GR GS IG

$$\text{IEC} = \frac{1}{N \times L}\sum_{i=1}^{N}\sum_{j=1}^{L} U_{i,j}^t \quad (6)$$

$\text{IEC}_{NR}$

$\hat{q} = [\ 3,\ 2,\ 1\ ]$
$q_k' = [\ 3,\ 2,\ 1\ ] \cdots K$
$\vdots$
$N \quad U_{i,j}^M = \begin{cases} 1 & \bar{r}_j^M = \bar{r}_j \\ 0 & \text{otherwise,} \end{cases} \quad (7)$

$\text{IEC}_{AR}$

$\hat{q} = [\ 0.6, 0.7, 0.2\ ]$
$q_k' = [\ 0.3, 0.5, 0.1\ ] \cdots K$
$\vdots$
$N \quad U_{i,j}^D = \begin{cases} 1 & \bar{Q}_{i,j}^D < \bar{Q}_{i,j} \\ 0 & \text{otherwise,} \end{cases} \quad (8)$

**Step 3. Integrating**

*Evaluate meta-consistency performance by combining the failure modes*

$$\text{MC} = \left(\frac{1}{|m^*|}\right) m^{*T} m \quad \text{where} \quad m = \begin{bmatrix} \text{IAC}_{NR} \\ \text{IAC}_{AR} \\ \text{IEC}_{NR} \\ \text{IEC}_{AR} \end{bmatrix} \quad (9)$$

NR / AR

$\text{IAC}_{AR}=0.90 \quad \text{IAC}_{NR}=0.08 \quad \text{IEC}_{AR}=0.54$
$\text{IEC}_{NR}=0.37$

Figure 4: Meta-evaluation of quality estimators is performed in three steps: (i) Perturbing, (ii) Scoring and (iii) Integrating. (i) First, a minor or disruptive perturbation is induced depending on the failure mode, i.e., $\mathcal{P}_\Omega^M$ for NR and $\mathcal{P}_\Omega^D$ for AR. (ii) Second, the estimator's intra- and inter-consistency are calculated to assess each performance dimension. The IAC score captures the extent that the estimator produces similar or dis-similar scores with respect to $\hat{q}$ and $q_k'$, which is illustrated through the distribution plots, where for NR and AR, the score distributions are overlapping and non-overlapping, respectively. The IEC score expresses ranking consistency. NR measures how consistently the estimator ranks different explanation methods and AR calculates how consistently the perturbed scores are lower than the unperturbed scores. (iii) In the final step, we integrate the previous steps and produce an MC score that summarises the estimator's performance: its resilience to noise and reactivity to adversary.

ranks different explanation methods similarly. We define $U^M$ for minor perturbations with entries such as:

$$U_{i,j}^M = \begin{cases} 1 & \bar{r}_j^M = \bar{r}_j \\ 0 & \text{otherwise,} \end{cases} \quad (7)$$

where $\bar{r}^M = r(\bar{Q}_{i,:}^M)$ with $\bar{Q}^M := \bar{Q}'$ and $\bar{r} = r(\bar{Q}_{i,:})$ are ranking vectors given a ranking measure $r : \mathbb{R}^L \mapsto \mathbb{R}^L$ that takes each row in $\bar{Q}_{i,:}^M$ and $\bar{Q}_{i,:}$, respectively and sorts the values in descending order. Each entry $\bar{r}_j^M \in \mathbb{N}$ corresponds to integers indicating their relative rank. For example, suppose we have one sample $x$, three explanation methods and their corresponding quality estimates, such as $\bar{Q}_{i,:}^M = [0.76, 0.86, 0.66]$. Then the results obtained from applying $r$ would be $\bar{r}^M = [2, 1, 3]$. An optimally-performing estimator would provide the same rankings for $\bar{r}^M$ as $\bar{r}$ for all $N$ inputs, resulting in IEC = 1. However, as discussed in Section 6, the reality is that many estimators often conflict with the optimal.

For disruptive perturbations, we interpret ranking consistency differently. Here, as explained in-depth in Appendix A.1.2, we measure how consistently the quality estimator ranks estimates from $\bar{Q}$ higher than $\bar{Q}^D := \bar{Q}'$. We define $U^D$ for disruptive perturbations with entries such as:

$$U_{i,j}^D = \begin{cases} 1 & \bar{Q}_{i,j}^D < \bar{Q}_{i,j} \\ 0 & \text{otherwise,} \end{cases} \quad (8)$$

where the quality estimates $\bar{Q}_{i,j}^D$ are generated for an explanation with respect to the same class as the one predicted for the unperturbed estimate $\bar{Q}_{i,j}$. For some estimators, e.g., in the robustness category, lower values are considered better than higher values, for which we invert the comparison symbol in Equation 8.

### 3.3 Quantifying Meta-Consistency

To conclude the framework, we want to characterise the performance of a quality estimator with a single Meta-Consistency (MC) score. To capture both the estimator's resilience to noise (NR) and its reactivity to

adversary (AR), we average over the two criteria for both failure modes:

$$\mathbf{MC} = \left(\frac{1}{|\boldsymbol{m}^*|}\right) \boldsymbol{m}^{*T} \boldsymbol{m} \quad \text{where} \quad \boldsymbol{m} = \begin{bmatrix} \mathbf{IAC}_{NR} \\ \mathbf{IAC}_{AR} \\ \mathbf{IEC}_{NR} \\ \mathbf{IEC}_{AR} \end{bmatrix} \tag{9}$$

and $\boldsymbol{m}^* = \mathbb{1}^4$ represents an optimally performing quality estimator as defined by the all-one indicator vector. A good quality estimator should produce an MC score close to 1 as higher values indicate better performance on the tested criteria[4], where $\mathbf{MC} \in [0, 1]$. An estimator that demonstrates a balance of resilience against minor perturbations and reactivity towards disruptive perturbations—as evidenced through its score distribution and ranking of different explanation methods—would achieve high meta-consistency scores with our framework. Our proposed score has the advantage of being both concise and comprehensive, as it provides a summary of the performance characteristics of an estimator while also taking into account multiple criteria. For a full overview of the framework, please see Figure 4.

## 4 Practical Evaluation

Within the framework of meta-evaluation, it is necessary to generate perturbed quality estimates for analysis. To accomplish this, we developed a series of practical tests. The methodology behind these tests is simple and thus easily extensible through the tests made available in the repository[5]. First, the space in which perturbations will be applied is selected, with options being either the input or the model. Second, based on the chosen space, an appropriate type of noise is defined. To ensure that the perturbations are meaningful and relevant to the task at hand, the noise type should be chosen contextually with respect to the data domain. For example, when perturbing the input space for images, we define a test as follows:

**Input Perturbation Test (IPT).** Apply i.i.d additive uniform noise such that $\hat{\boldsymbol{x}}_i = \boldsymbol{x} + \boldsymbol{\delta}_i$ with $\boldsymbol{\delta}_i \sim \mathcal{U}(\alpha, \beta)$ where for noise resilience, $\hat{\boldsymbol{x}}_i$ fulfills Definition 1 and for adversary reactivity, $\hat{\boldsymbol{x}}_i$ fulfills Definition 2

where $\alpha, \beta$ have to be chosen according to the data domain and respective failure mode (e.g., set $\alpha = -0.001, \beta = 0.001$ for NR and $\alpha = 0.0, \beta = 1.0$ for AR). To maintain the statistics of the data distribution, we clip $\alpha, \beta$ to the maximum and the minimum value of the test set, respectively. Moreover, when perturbing the model space, to maintain the variance of the network, we follow an established methodology by Bykov et al. (2022) and scale the learned weights $\boldsymbol{\theta}$ of the model $f$ as follows:

**Model Perturbation Test (MPT).** Apply multiplicative Gaussian noise to all weights of the network, i.e., $\hat{\boldsymbol{\theta}}_i = \boldsymbol{\theta} \cdot \boldsymbol{\nu}_i$ with $\boldsymbol{\nu}_i \sim \mathcal{N}(\boldsymbol{\mu}, \boldsymbol{\Sigma})$ where $\boldsymbol{\mu} = 1$ and for noise resilience, $\hat{\boldsymbol{\theta}}_i$ fulfills Definition 1 and for adversary reactivity, $\hat{\boldsymbol{\theta}}_i$ fulfills Definition 2

where for $\boldsymbol{\Sigma}$ to be consistent with either Definition 1 or 2, it is set based on the specific context of the model and task being considered (e.g., $\boldsymbol{\Sigma} = 0.001$ for NR and $\boldsymbol{\Sigma} = 2.0$ for AR). Third, to collect sets of perturbed quality estimates for intra- and inter-consistency analysis, we repeat the process of perturbation (as outlined in IPT and MPT) and subsequent evaluation (using Equation 4) under $K$ runs. Finally, we compute the MC score. For sanity-checking experiments of the tests, see Appendix A.5. Moreover, as a third testing scenario, it is theoretically possible to perturb both the input- and model spaces simultaneously, i.e., $\mathcal{P}_{\mathbb{X}}(\boldsymbol{x}), \mathcal{P}_{\mathbb{F}}(\theta)$ as well as their respective latent spaces. This we leave for future work.

## 5 Experimental Setup

In this section, we give a brief account of the experimental setup, including the datasets, models, explanation methods and estimators used in this work. Further details can be found in Appendix A.4.

---

[4]When computing intra-consistency scores for AR, we apply reverse scoring, i.e., $1 - \mathbf{IAC}_{AR}$, so that all elements in the meta-evaluation vector (Equation 9) can be interpreted in the same way, i.e., that higher values are better.

[5]Code is available at the GitHub repository: https://github.com/annahedstroem/MetaQuantus.

In our experiments, we benchmark five different categories of explanation quality and within each category, we selected two estimators as follows: *Complexity* (CO) (Bhatt et al., 2020), *Sparseness* (SP) (Chalasani et al., 2020), *Faithfulness Correlation* (FC) (Bhatt et al., 2020), *Pixel-Flipping* (PF) (Bach et al., 2015), *Max-Sensitivity* (MS) (Yeh et al., 2019), *Local Lipschitz Estimate* (LLE) (Alvarez-Melis & Jaakkola, 2018), *Pointing-Game* (PG) (Zhang et al., 2018), *Relevance Mass Accuracy* (RMA) (Arras et al., 2022), *Model Parameter Randomisation Test* (MPR) (Adebayo et al., 2018) and *Random Logit* (RL) (Sixt et al., 2020). Each estimator evaluates explanations from a popular category of post-hoc attribution methods, including both gradient-based- and model-agnostic techniques: *Gradient* (Morch et al., 1995; Baehrens et al., 2010), *Saliency* (Morch et al., 1995), *GradCAM* (Selvaraju et al., 2020), *Integrated Gradients* (Sundararajan et al., 2017), *Input×Gradient* (Shrikumar et al., 2016), *Occlusion* (Zeiler & Fergus, 2014) and *GradientSHAP* (Lundberg & Lee, 2017) from which we generate explanations with respect to a sample's predicted class. For comparability, we normalise the explanations by dividing the attribution map by the square root of its average second-moment estimate (Binder et al., 2022). The mathematical definitions of the estimators are described in Appendix A.3. For estimator implementations, we use `Quantus` library (Hedström et al., 2023).

We use four image classification datasets for our experiments: ILSVRC-15 (i.e., ImageNet) (Russakovsky et al., 2015), MNIST (LeCun et al., 2010), fMNIST (Xiao et al., 2017) and customised-MNIST (i.e., cMINST) (Bykov et al., 2022) with different black-box NNs such as LeNets (LeCun et al., 1998), ResNets (He et al., 2016) and transformer-based models such as Vision Transformer (ViT) (Dosovitskiy et al., 2021) and SWIN Transformer (Liu et al., 2021b).

## 6 Results

Many open questions remain in the field of XAI. In this section, we show how meta-evaluation can solve a subset of those problems such as (i) estimator selection, (ii) optimising hyperparameters of an estimator, (iii) evaluating the category convergence, i.e., the extent that estimators within the same category of explanation quality measure the same concept and (iv) comparing estimator choice for different network architectures including transformer-based architectures. We prioritise the topic of estimator selection in the main manuscript and provide a detailed analysis and discussion of the experiments addressing questions (ii), (iii) and (iv) in Appendix A.6. Instructions to reproduce the experiments can be found in the repository.

Table 1: Meta-evaluation benchmarking results with ImageNet, aggregated over 3 iterations with $K = 5$. IPT results are in grey rows and MPT results are in white rows. $\overline{\text{MC}}$ denotes the averages of the MC scores over IPT and MPT. The top-performing MC- or $\overline{\text{MC}}$ method in each explanation category, which outperforms the bottom-performing method by at least 2 standard deviations, is underlined. Higher values are preferred for all scoring criteria.

| Category | Estimator | $\overline{\text{MC}}$ (↑) | MC (↑) | $\text{IAC}_{NR}$ (↑) | $\text{IAC}_{AR}$ (↑) | $\text{IEC}_{NR}$ (↑) | $\text{IEC}_{AR}$ (↑) |
|---|---|---|---|---|---|---|---|
| *Complexity* | Sparseness | $0.636 \pm 0.055$ | $0.697 \pm 0.048$ | $0.519 \pm 0.229$ | $0.745 \pm 0.145$ | $0.870 \pm 0.060$ | $0.653 \pm 0.064$ |
| | | | $0.575 \pm 0.062$ | $0.588 \pm 0.171$ | $0.445 \pm 0.164$ | $0.862 \pm 0.047$ | $0.403 \pm 0.036$ |
| | Complexity | $0.562 \pm 0.033$ | $0.590 \pm 0.043$ | $0.170 \pm 0.093$ | $0.965 \pm 0.092$ | $1.000 \pm 0.000$ | $0.225 \pm 0.068$ |
| | | | $0.534 \pm 0.023$ | $0.221 \pm 0.086$ | $0.792 \pm 0.088$ | $1.000 \pm 0.000$ | $0.122 \pm 0.012$ |
| *Faithfulness* | Faithfulness Corr. | $0.480 \pm 0.080$ | $0.470 \pm 0.078$ | $0.518 \pm 0.241$ | $0.550 \pm 0.138$ | $0.342 \pm 0.063$ | $0.470 \pm 0.051$ |
| | | | $0.490 \pm 0.082$ | $0.497 \pm 0.216$ | $0.586 \pm 0.149$ | $0.329 \pm 0.074$ | $0.546 \pm 0.039$ |
| | Pixel-Flipping | $\underline{0.698 \pm 0.095}$ | $0.623 \pm 0.128$ | $0.518 \pm 0.232$ | $0.691 \pm 0.405$ | $0.707 \pm 0.052$ | $0.574 \pm 0.125$ |
| | | | $0.774 \pm 0.062$ | $0.515 \pm 0.224$ | $1.000 \pm 0.000$ | $0.719 \pm 0.060$ | $0.863 \pm 0.011$ |
| *Localisation* | Pointing-Game | $0.537 \pm 0.043$ | $0.591 \pm 0.066$ | $0.685 \pm 0.121$ | $0.579 \pm 0.159$ | $0.903 \pm 0.049$ | $0.196 \pm 0.032$ |
| | | | $0.483 \pm 0.020$ | $0.868 \pm 0.068$ | $0.102 \pm 0.098$ | $0.935 \pm 0.026$ | $0.026 \pm 0.012$ |
| | Relevance Mass Acc. | $0.580 \pm 0.070$ | $0.619 \pm 0.067$ | $0.524 \pm 0.213$ | $0.633 \pm 0.172$ | $0.750 \pm 0.044$ | $0.570 \pm 0.047$ |
| | | | $0.541 \pm 0.074$ | $0.558 \pm 0.125$ | $0.476 \pm 0.199$ | $0.767 \pm 0.052$ | $0.364 \pm 0.037$ |
| *Randomisation* | Random Logit | $0.604 \pm 0.032$ | $0.644 \pm 0.063$ | $0.481 \pm 0.268$ | $0.631 \pm 0.217$ | $0.939 \pm 0.035$ | $0.524 \pm 0.083$ |
| | | | $0.564 \pm 0.000$ | $0.393 \pm 0.000$ | $0.400 \pm 0.000$ | $0.960 \pm 0.000$ | $0.505 \pm 0.000$ |
| | Model Param. Rand. | $\underline{0.697 \pm 0.042}$ | $0.762 \pm 0.041$ | $0.454 \pm 0.170$ | $0.909 \pm 0.000$ | $0.967 \pm 0.020$ | $0.720 \pm 0.022$ |
| | | | $0.631 \pm 0.044$ | $0.465 \pm 0.200$ | $0.563 \pm 0.077$ | $0.954 \pm 0.017$ | $0.541 \pm 0.016$ |
| *Robustness* | Max-Sensitivity | $0.610 \pm 0.065$ | $0.621 \pm 0.068$ | $0.507 \pm 0.144$ | $0.593 \pm 0.284$ | $0.938 \pm 0.026$ | $0.446 \pm 0.089$ |
| | | | $0.600 \pm 0.061$ | $0.454 \pm 0.251$ | $1.000 \pm 0.000$ | $0.921 \pm 0.020$ | $0.023 \pm 0.007$ |
| | Local Lipschitz Est. | $0.615 \pm 0.071$ | $0.635 \pm 0.112$ | $0.565 \pm 0.099$ | $0.630 \pm 0.561$ | $0.916 \pm 0.031$ | $0.431 \pm 0.197$ |
| | | | $0.594 \pm 0.030$ | $0.498 \pm 0.144$ | $0.485 \pm 0.073$ | $0.895 \pm 0.027$ | $0.498 \pm 0.016$ |

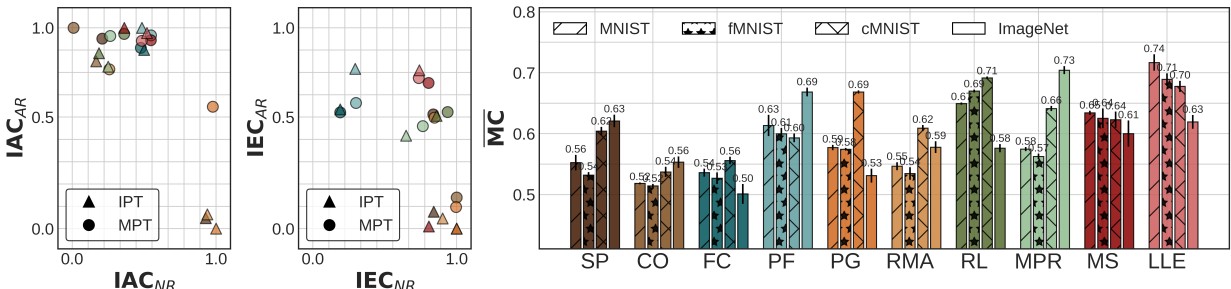

Figure 5: *Left:* A visualisation of MNIST benchmarking results (Table 3), in particular IAC and IEC scores for noise resilience (x-axes) and adverse reactivity (y-axes). The colours indicate the estimator and the symbols show the test type, i.e., IPT and MPT, respectively. *Right:* A comparison of averaged meta-consistency performance for different quality estimators using MPT and IPT, aggregated over 3 iterations with $K = 5$, across different models {*LeNets, ResNet-9, ResNet-18*} and datasets {*MNIST, fMNIST, cMNIST, ImageNet*}. Higher values are preferred.

## 6.1 Benchmarking

As a first example, we will demonstrate how meta-evaluation can be used to select a certain quality estimator for a given category of explanation quality. To this end, we set up a benchmarking experiment, where we take two popular estimators from five different explanation quality categories and evaluate six explanation methods $L$ ={*Gradient, Saliency, GradCAM, Integrated Gradients, Occlusion, GradientShap*} for MNIST, fMNIST and cMNIST datasets and evaluate three explanation methods $L$ ={*Saliency, Input×Gradient, GradientShap*} for ImageNet dataset, using the `MetaQuantus` library. Since the choice of $L$ has a minimal influence on the MC scores (see experiments in Appendix A.5.2), we omit results from other tested sets of explanation methods in the main manuscript.

### 6.1.1 Comparison of XAI Estimators

The ImageNet results are summarised in Table 1. The grey rows indicate the results from the Input Perturbation Test and the white rows show the results from the Model Perturbation Test. A more detailed discussion of the results, including additional datasets, can be found in Appendix A.6.3. From Table 1, we can observe that no tested estimator performs optimally, i.e., $\forall$ MC $< 1$. From column $\overline{\text{MC}}$, which displays the averaged MC scores (over IPT and MPT) we note that *Sparseness*, *Pixel-Flipping*, *Relevance Mass Accuracy*, *Model Parameter Randomisation Test* and *Local Lipschitz Estimate* are the best-performing estimators in their respective category for ImageNet dataset. From Figure 5 (right), we can observe that this comparison of MC scores is generally consistent across the tested datasets, which contributes to the generalisability of our findings. Certain categories such as localisation and randomisation exhibit higher variability with respect to the MC score, which we further discuss in Appendix A.6.5.

### 6.1.2 Comparison of XAI Evaluation Categories

The meta-evaluation framework can moreover be applied to gain insights into the performance characteristics of different estimators on a category-by-category basis. For this purpose, we represent the entries of the meta-evaluation vector as coordinates on a 2D plane and visualise the results as an area graph (see Figure 7). By inspecting the coloured areas of the respective estimators in terms of their size and shape, we can deduce the overall performance of both failure modes. Here, larger coloured areas imply better performance on the different scoring criteria and the grey area indicates the area of an optimally performing quality estimator, i.e., $\boldsymbol{m}^* = \mathbb{1}^4$. Each column of estimators represents a category of explanation quality, from left to right: *Complexity*, *Faithfulness*, *Localisation*, *Randomisation* and *Robustness*, which colour scheme we apply consistently across all figures.

As seen in Figure 7 (third column), the localisation estimators exhibit a notable deficiency in terms of adversary reactivity on the Input Perturbation Test. A low IPT score for adversary reactivity means that the estimators are insensitive to disruptive input perturbations, as evidenced by similar score distributions

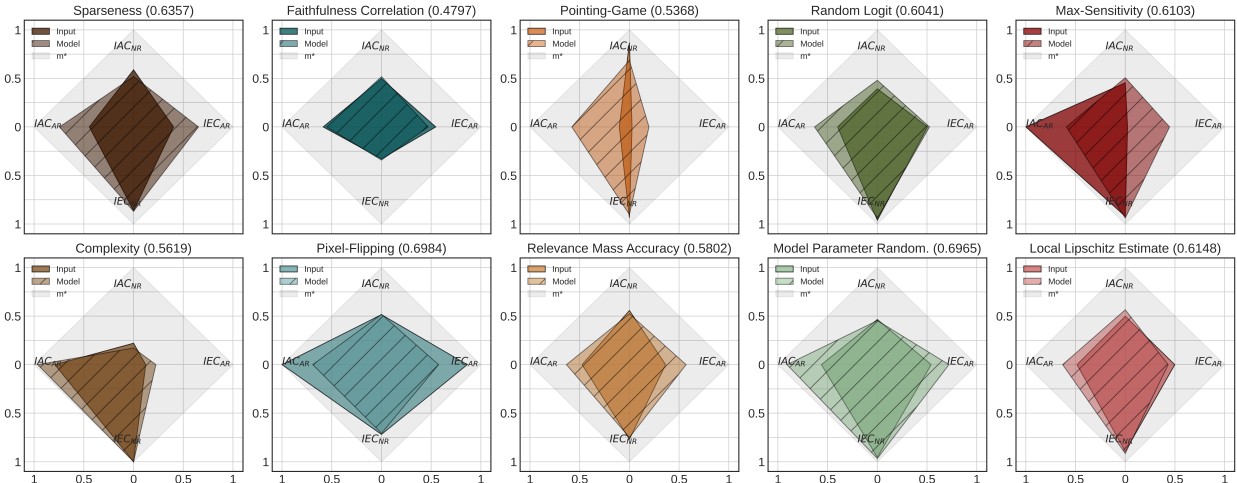

Figure 7: A graphical representation of the ImageNet benchmarking results (Table 1), aggregated over 3 iterations with $K = 5$. Each column corresponds to a category of explanation quality, from left to right: *Complexity*, *Faithfulness*, *Localisation*, *Randomisation* and *Robustness*. The grey area indicates the area of an optimally performing estimator, i.e., $\mathbf{m}^* = \mathbb{1}^4$. The MC score (indicated in brackets) is averaged over MPT and IPT. Higher values are preferred.

(low IAC) and an inability to rank disruptively perturbed explanations lower than unperturbed explanations (low IEC). Based on the definitions of these estimators (described in Equations 24-25), which include the *Pointing-Game* method (Zhang et al., 2018), which evaluates explanations by verifying that the highest attributed feature intersects with a given segmentation mask and the *Relevance Mass Accuracy* method (Arras et al., 2022), which calculates the amount of explainable mass intersecting with the segmentation mask—we would expect that these estimators perform well on this test since disrupted input usually leads to scattered attributions. It may seem counterintuitive that these estimators lack reactivity to disruption, however, we posit that the reason for the poor reactivity to adversary is the estimators' inherent dependency on the segmentation mask. If the segmentation mask (relative to the object of interest, or the input) is large enough, high localisation scores are attainable irrespective of the "quality" of the explanation (Kohlbrenner et al., 2020). This finding is further validated by the increase in MC scores for cMNIST dataset, which generally has a smaller bounding box compared to ImageNet, MNIST and fMNIST (see details in Appendix A.4), where disruption evidently has an effect, as evidenced by higher AR scores, depicted in Figure 5 (right). Practitioners should be aware of this category's reliance (or oversensitivity) on the segmentation mask where relying solely or too heavily on this category in XAI evaluation may not be advisable.

The highest overall meta-consistency scores are obtained by the robustness and randomisation categories, which can be observed in Figure 6 and by their respective areas in Figure 7. One potential explanation for this is that the estimators in these categories already include some element of stochasticity in their estimator definitions (see Equations 18-19 and 20-21, respectively) which may make them more resilient as well as reactive to perturbations. For example, both robustness estimators, i.e., *Max-Sensitivity* (Yeh et al., 2019) and *Local Lipschitz Estimate* (Alvarez-Melis & Jaakkola, 2018) rely on Monte Carlo sampling-based approximation where explanation methods are evaluated by examining their response to minor perturbation of the input, aggregated over multiple runs. In the randomisation category, the *Model Parameter Randomisation Test* (Adebayo et al., 2018) evaluates explanations by increasingly perturbing the model weights and *Random Logit* (Sixt et al., 2020) evaluates explanations by a random selection of an explanation of a non-target class. As seen in Figure 6, the complexity category has the lowest overall MC scores across the tested datasets, which includes estimators such as *Sparseness* (Chalasani et al.,

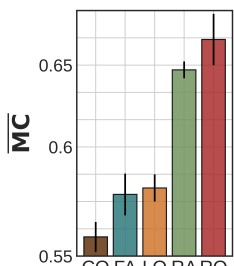

Figure 6: Average $\overline{\mathrm{MC}}$ scores per category over tested datasets {*ImageNet*, *MNIST*, *fMNIST*, *cMNIST*}.

2020) and *Complexity* (Bhatt et al., 2020) that evaluate explanations by calculating their Gini coefficient and Shannon entropy, respectively. Given the simplicity of these calculations, this outcome is not surprising.

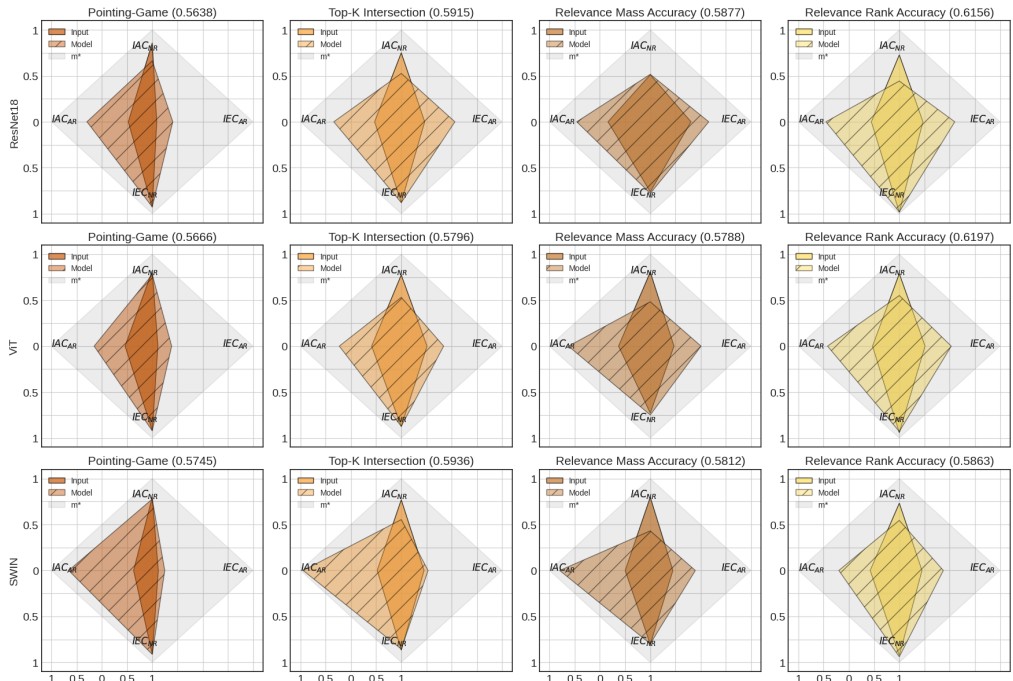

Figure 8: A graphical representation of the ImageNet benchmarking results for different architectures aggregated over 3 iterations with $K = 5$. Each column represents the estimator, from left to right: *Pointing-Game*, *Top-K Intersection*, *Relevance Mass Accuracy* and *Relevance Rank Accuracy*, and each row the model type {*ResNet-18*, *ViT*, *SWIN*}, as indicated by the labels. The grey area indicates the area of an optimally performing estimator, i.e., $\mathbf{m}^* = \mathbb{1}^4$. The MC score (indicated in brackets) is averaged over MPT and IPT. Higher values are preferred.

Another notable category of poor performance that is picked up by the meta-evaluation tests is faithfulness. Our results, which show a lack of resilience to noise in the ranking of explanation methods (low IEC) especially for *Faithfulness Correlation* (Bhatt et al., 2020), corroborate previous studies (Brunke et al., 2020; Brocki & Chung, 2022; Rong et al., 2022) that found that faithfulness estimators may rank explanation methods inconsistently when subjected to perturbation, such as changing the masking pixel strategy from, for example, "uniform" to "black". This trend is particularly evident in Figure 5 (left), where the points belonging to the faithfulness category have notably lower IEC scores compared to the other categories of explanation quality. A possible explanation is their well-documented sensitivity to parameterisation (Tomsett et al., 2020; Hedström et al., 2023; Rong et al., 2022).

While certain estimator categories, such as faithfulness, may present challenges such as parameterisation, it is not advisable to disregard their evaluation. Compared to categories such as complexity which are well-defined and simple to calculate, they may not offer as much information as categories such as faithfulness, which can provide important insights into how the explanation- and model functions are related. Relying on only one category to estimate explanation quality is therefore not recommended. This is especially true since an explanation function may trade one category of explanation quality over another (Bhatt et al., 2020), for example, an explanation that is faithful may be too complex for the user to understand. Therefore, to avoid arriving at incomplete or incorrect conclusions about which explanation methods work (and not), it is of utmost importance for practitioners to approach quality estimation by incorporating multiple definitions of explanation quality.

### 6.1.3   Comparison of SOTA Vision Architectures

To investigate if the performance characteristics of the different quality estimators revealed by the meta-evaluation endure across various architectural types, we conducted additional experiments with ResNet- and transformer-based model architectures. These architectures included the Vision Transformer (ViT) (Dosovitskiy et al., 2021) and the SWIN Transformer (Liu et al., 2021b). For this experiment, we employed two

explanation methods $L = \{$*Saliency, GradientShap*$\}$ and meta-evaluated four estimators in the localisation category, namely *Pointing-Game* (PG) (Zhang et al., 2018), *Relevance Mass Accuracy* (RMA) (Arras et al., 2022), *Relevance Rank Accuracy* (RRA) (Arras et al., 2022) and *Top-K Intersection* (TK) (Theiner et al., 2022). These estimators are all defined in Appendix A.3. We also conducted a comparable experiment with the complexity estimators *Sparseness* (Chalasani et al., 2020) and *Complexity* (Bhatt et al., 2020). These results are found in Appendix A.6.4, which yielded outcomes that are consistent with the findings presented here.

In Figure 8, we present the meta-evaluation results as area graphs. Each column represents a localisation estimator and each row represents a network architecture, as indicated by the labels. The performance attributes of each estimator, when examined across different network types, display a noticeable similarity. This suggests that the characteristics revealed through meta-evaluation analysis may be persistent across architecture types, thereby enabling general conclusions of individual estimators. In particular, *Pointing-Game* (Zhang et al., 2018) performed poorly across all model architectures, with low adverse reactivity in the rankings (IEC) where *Relevance Mass Accuracy* (Arras et al., 2022) generally demonstrated higher performance. *Relevance Rank Accuracy* (Arras et al., 2022) and *Top-K Intersection* (Theiner et al., 2022) showed the highest scores with narrow margins, possibly due to their similarity in definitions (see Equations 26 and 27 in Appendix A.3). Overall, these results contribute to the robustness of our findings and provide evidence demonstrating the framework's agnosticism to the choice of network architecture.

# 7 Conclusion

When we neither understand the general behaviour of the explanation methods nor the estimators that we apply to estimate their quality, we are bound to make mistakes. This problem in XAI is exacerbated by the fact that different estimators within the same category of explanation quality may rank the same explanation method differently (Neely et al., 2021; Krishna et al., 2022). Without an understanding of the performance characteristics of the estimators we employ, we risk presenting inferior explanation methods to the end user.

To address the problem of meta-evaluation, we propose a novel framework for identifying reliable quality estimators to evaluate explanation methods. We circumvent the *Challenge of Unverifiability* by evaluating the estimators through the lens of reliability—through perturbing the verifiable variables of XAI evaluation and thereafter analysing the estimator's outcomes under different failure modes, we can get an objective and independent characterisation of its performance. We show, in a series of experiments, how to use the framework for estimator selection and for systematic meta-evaluation of the strengths and weaknesses of individual estimators (Section 6.1.1), general categories of explanation quality (Section 6.1.2) as well as architecture types (Section 6.1.3). Our findings show that (i) localisation estimators demonstrate a deficiency in terms of adversary reactivity, possibly due to their dependency on the segmentation mask, (ii) faithfulness estimators are inconsistent in their ranking and (iii) estimators within the randomisation- and robustness categories demonstrate the highest performance. It is advised that XAI practitioners take into account the limitations of various estimator categories and exercise caution when relying heavily on certain categories.

Evaluating the intrinsic value or "validity" of a quality estimator is, however, still an open and important question to consider. It is essential to keep in mind that the reliability of an estimator does not necessarily imply any intrinsic validity, e.g., an estimator's theoretical soundness (Sundararajan et al., 2017; Binder et al., 2022). As we make the meta-evaluation tests readily available, we inadvertently create the risk of the tests being misused and the results being misinterpreted. We encourage the reader to exercise caution when interpreting the outcome of the tests and not rush to conclusions regarding the overall reliability of quality estimators, which should always be judged with respect to their specific experimental context.

Our experiments are limited to image classification applications. To fully demonstrate the generalisability of `MetaQuantus`, we plan to extend our experiments more broadly in the sciences and medicine and include other explanation method types as well as other data domains such as tabular, textual and time series data. This will require additional work to ensure that the estimators themselves support these data domains, which will be addressed in upcoming publications.

**Acknowledgments**

This work was partly funded by the German Federal Ministry for Education and Research through project Explaining 4.0 (ref. 01IS200551), BIFOLD (ref. 01IS18025A and ref. 01IS18037A), the Investitionsbank Berlin through BerDiBa (grant no. 10174498), the European Union's Horizon 2020 research and innovation programme through iToBoS (grant no. 965221) as well as European Union's Horizon 2022 research and innovation programme (EU Horizon Europe) TEMA (grant no. 101093003). It was also financially supported by the Research Council of Norway (RCN), through the FRIPRO (grant no. 315029), and partially funded by its Centre for Research-based Innovation funding scheme (Visual Intelligence, grant no. 309439) and Consortium Partners, RCN IKTPLUSS (grant no. 303514), and the UiT Thematic Initiative "Data-Driven Health Technology".

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

# A    Appendix

In this section, we include all the necessary details and information to support the claims and results presented in the main paper. In Section A.1, we present theoretical considerations for the meta-evaluation framework. In Sections A.2 and A.3, we outline the mathematical definitions of explanation quality, for categories and estimators, respectively. In Section A.4, we describe the experimental setup and in the following Section A.5, we describe the experiments that were performed to sanity-check the framework. In Section A.6, we provide supplementary results, in terms of additional applications and supporting experiments. We provide a notation table at the end in Section A.7.

### Broader Impact Statement

This research is important since we raise awareness and address the need for reliable evaluation methods in the Explainable AI (XAI) community. In the XAI community, the evaluation of explanation methods has often been neglected or clouded by the ambiguity that an absence of ground truth labels entails—yet to foster sustainable progress in the field over time it is necessary to systematically define and meta-evaluate the methods used to measure the quality of explanations. This research takes the first step towards this goal by developing practical, quantifiable tools for reliable XAI evaluation. Without careful examination of the quality of explanations, the deployment of potentially beneficial machine learning algorithms may be hindered, preventing the full potential of AI from being realised in important areas such as healthcare, education, finance and policy.

## A.1    Theoretical Considerations

This section discusses the concept of minor and disruptive perturbations in the context of multi-label classification and regression tasks in XAI. It also addresses the potential vulnerability of the framework to adversarial attacks and the motivation for the calculation of the IEC scoring criterion for disruptive perturbations.

### A.1.1    Expanding Scope and Adversarial Attacks

**Multi-label classification**    In this work, given the popularity of image classification tasks in XAI, we focused mostly on this application. In this application, the definitions of minor and disruptive perturbations (as given in Definitions 1 and 2) apply to $\hat{y}$ and $y'$ which represent the predicted classes for true and perturbed samples, respectively. However, our definitions can be easily extended to other types of classification tasks, such as multi-label classification. In this case, for a multi-label classification problem with $C$ classes, the definitions of minor and disruptive perturbations (as given in Definitions 1 and 2) would apply to binary prediction vectors $\hat{\boldsymbol{y}} \in \mathbb{R}^C$ and $\boldsymbol{y}' \in \mathbb{R}^C$, rather than single classes. The distance between the two vectors can be denoted using the $|||\cdot|||_n$ notation.

**Regression**    As is the case with many explanation methods (Letzgus et al., 2021), the extension to the regression problem in XAI is not straightforward. Given $y$ and $y'$ as real-valued prediction outcomes, we would need to adjust Definitions 1 and 2 to encompass a derivation of a proper boundary $\epsilon$. We leave the task of adapting the meta-evaluation framework to regression problems to future work.

**Adversarial attacks**    Adversarial attacks are techniques used to manipulate or deceive models (Szegedy et al., 2014) or their explanations (Dombrowski et al., 2019) by introducing perturbations to the input data that are imperceptible to humans but results in an incorrect prediction by the model. To adversarially attack Definition 1, it is theoretically possible to define a perturbation that maximises the strength of the perturbation, while still remaining consistent with Definition 1, i.e., $||\hat{y} - y'||_p > \epsilon$. In the same vein, to attack Definition 2, it is theoretically possible to define a perturbation that minimises the strength of the perturbation, while still remaining consistent with Definition 2, i.e., $||\hat{y} - y'||_p \leq \epsilon$. While it may be a theoretical possibility to attack the framework through the definitions, performing such attacks would not serve any practical purpose. This is because it contradicts the purpose of the framework, which is to assist practitioners in selecting and developing reliable XAI methods.

### A.1.2 Motivation for IEC$_{\text{AR}}$ calculation

Contrary to the calculation of the inter-consistency criterion for minor perturbations, i.e., IEC$_{\text{NR}}$, we cannot motivate IEC$_{\text{AR}}$ based on ranking consistency with respect to explanation methods, as disruptive perturbations implicate a change in the estimator score. While the expectation of changed rankings as a result of disruptive perturbations may appear intuitive, the imposed change on the quality estimators could theoretically lead to a symmetrical change across explanation method scores, which preserves the ranking across explanation methods. Since the behaviour of the explanation functions under disruptive perturbations lies in the unverifiable spaces $\mathbb{U}$, we cannot exclude the possibility of a symmetrical response. Accordingly, for the calculation of IEC$_{AR}$, we relax the theoretical assumptions to a ranking comparison based on scores (as defined in Equation 7) which remain in the verifiable spaces $\mathbb{\Omega}$.

The assumption for the calculation of the IEC score with respect to disruptive perturbation is motivated by the scenario of an ideal estimator, which is expected to be able to assess the true performance of an explanation method, denoted $q^{\text{true}}$. In the ideal scenario, the real performance varies only slightly, i.e., $q_j^{\text{true}} \pm \epsilon$ would therefore define an upper estimation bound $q_j^{\text{true}} \approx q_j^{\text{max}}$ for each explanation method $j \in [1, \ldots, L]$[6]. All estimates $\bar{Q}_{i,j}^D$ resulting from the AR scenario should differ from the unperturbed quality estimate $\bar{Q}_{i,j}^D \neq \bar{Q}_{i,j}^*$. In the idealised scenario $q_j^{\text{true}} \approx q_j^{\text{max}}$, we argue that $\bar{Q}_{i,j}^* \approx q_j^{\text{true}}$ and $\bar{Q}_{i,j}^D < \bar{Q}_{i,j}^*$, leading to Equation 8. Note, however, that in practice quality estimates are subject to larger variations which means that the assumption $q_j^{\text{true}} \approx q_j^{\text{max}}$ and Equation 8 might not always hold. Therefore, in practice, we do not expect IEC$_{AR} \approx 1$, which aligns with our results in Table 1. Nonetheless, further research on the inter-consistency criterion under disruptive perturbations is subject to future work.

### A.2 Explanation Quality: Category Definitions

In the main paper, we described how a lack of explanation ground truth labels has led to a diverse set of interpretations of explanation quality. In the following, we provide a brief summary of the most established categories of explanation quality, grouped into six categories; (a) faithfulness, (b) robustness, (c) randomisation, (d) complexity, (e) localisation and (f) axiomatic estimators (Hedström et al., 2023). To establish a mathematical ground for each category, we present a summarising equation. This means that all the nuances that typically exist within a category of explanation quality are not considered. For completeness, we, therefore, provide the exact mathematical descriptions of each of the individual estimators used in this work in Appendix A.3.

Since many explanation categories do rely on perturbation, we define a general perturbation function on any real-valued space $\mathbb{S} \subseteq \mathbb{R}^N, N \in \mathbb{N}$ in the following.

**Definition 3** (Perturbation). Let $\mathcal{P}_{\mathbb{S}}(\boldsymbol{s}; \eta) : \mathbb{S} \mapsto \mathbb{S}$ be a perturbation function of $\boldsymbol{s} \in \mathbb{S}$ with parameters $\eta \in \mathbb{R}$ such that $\forall \hat{\boldsymbol{s}} \in \mathbb{S}, \hat{\boldsymbol{s}} \neq \boldsymbol{s}$:

$$\mathcal{P}_{\mathbb{S}}(\boldsymbol{s}; \eta) = \hat{\boldsymbol{s}}. \tag{10}$$

For simplicity, we also write $\mathcal{P}_{\mathbb{S}}(\boldsymbol{s}) := \mathcal{P}_{\mathbb{S}}(\boldsymbol{s}; \eta)$, which is used in the main paper.

**Faithfulness** (Montavon et al., 2018; Samek et al., 2017; Bach et al., 2015; Bhatt et al., 2020; Nguyen & Martinez, 2020) quantifies the extent that explanations follow the predictive behaviour of the model, asserting that more important features affect model decisions more strongly. Given $f$, $\boldsymbol{x}$, $y'$ and $\hat{\boldsymbol{e}}$, the change in the model output $f(\boldsymbol{x})$ is measured as the input features of $\boldsymbol{x}$ are manipulated based on their attribution in $\hat{\boldsymbol{e}}$. The input manipulation is defined as a perturbation function $\mathcal{P}_{\mathbb{X}}(\boldsymbol{x}, M)$ with $\boldsymbol{x} \in \mathbb{X}$ where $M$ is the number of input features that are perturbed. Since $f$ denotes a trained model with parameters $\theta$, for brevity, we denote $f(\boldsymbol{x}; \theta)$ as $f(\boldsymbol{x})$ where possible.

$$\Psi_{\text{F}}(\Phi, f, \boldsymbol{x}, \mathcal{P}, M) = |(f(x) - f(\mathcal{P}_{\mathbb{X}}(\boldsymbol{x}, M))|. \tag{11}$$

---

[6]Here, we present general theoretical considerations, but the specific claims for each estimator would require individual proofs.

**Robustness** ([Montavon et al., 2018](#); [Alvarez-Melis & Jaakkola, 2018](#); [Yeh et al., 2019](#); [Dasgupta et al., 2022](#)) measures the stability of the explanation function with respect to small changes in the input, requiring that those small perturbations in the input space $||\mathcal{P}_{\mathbb{X}}(\boldsymbol{x}) - \boldsymbol{x}||_p < \varepsilon$, e.g., under an $\ell_p$ norm constraint upper bounded by some positive constant $\varepsilon$, lead to only slight changes in the explanation $||\hat{\boldsymbol{e}} - \Phi(\mathcal{P}_{\mathbb{X}}(\boldsymbol{x}), f, \hat{y})|| < \varepsilon$ assuming that the model output approximately stayed the same $f(\boldsymbol{x}) \approx f(\mathcal{P}_{\mathbb{X}}(\boldsymbol{x}))$.

$$\Psi_{\text{RO}}(\Phi, f, \boldsymbol{x}, \hat{y}, \hat{\boldsymbol{e}}, \mathcal{P}) = ||\hat{\boldsymbol{e}} - \Phi(\mathcal{P}_{\mathbb{X}}(\boldsymbol{x}), f, \hat{y}; \lambda)|| \leq \varepsilon. \tag{12}$$

**Randomisation** ([Adebayo et al., 2018](#); [Sixt et al., 2020](#)) measures the extent explanations deteriorate as randomness is introduced to the quality estimation. For example, [Adebayo et al. (2018)](#) measure the change in explanation as model parameters $\theta$ are increasingly randomised, requiring large perturbations in the parameter space of the model, i.e., $\mathcal{P}_{\mathbb{F}}(\theta) \gg \varepsilon$ to result in large changes in the explanation, i.e., $||\hat{\boldsymbol{e}} - \Phi(\boldsymbol{x}, \mathcal{P}_{\mathbb{F}}(\theta), \hat{y}; \lambda)|| \gg \varepsilon$.

$$\Psi_{\text{RA}}(\Phi, f, \boldsymbol{x}, \hat{y}, \hat{\boldsymbol{e}}, \mathcal{P}, \varepsilon) = ||\hat{\boldsymbol{e}} - \Phi(\boldsymbol{x}, \mathcal{P}_{\mathbb{F}}(\theta), \hat{y}; \lambda)|| \gg \varepsilon. \tag{13}$$

**Complexity** ([Bhatt et al., 2020](#); [Chalasani et al., 2020](#); [Nguyen & Martinez, 2020](#)) captures the conciseness of explanations, i.e., only a few features should be selected to explain a model prediction. The notion of complexity differs in how it is empirically interpreted, e.g., by computing the Shannon entropy of attribution map ([Bhatt et al., 2020](#)). Alternatively, [Chalasani et al. (2020)](#) quantifies complexity by calculating the Gini Index of the absolute value of the attribution vector $\hat{\boldsymbol{e}}$ where $D$ is the length of the attribution vector.

$$\Psi_{\text{C}}(\hat{\boldsymbol{e}}) = \frac{\sum_{i=1}^{D}(2i - D - 1)\hat{\boldsymbol{e}}_i}{D \sum_{i=1}^{D} \hat{\boldsymbol{e}}_i}. \tag{14}$$

**Localisation** ([Theiner et al., 2022](#); [Kohlbrenner et al., 2020](#); [Zhang et al., 2018](#); [Rong et al., 2022](#); [Arias-Duart et al., 2021](#)) tests if the explainable evidence is centred around a region of interest, which may be defined around an object by a bounding box, a segmentation mask or a cell within a grid. It requires an additional segmentation mask $\boldsymbol{s}^{gt} \in \mathbb{R}^D$, mostly a binary mask of the input $\boldsymbol{s}_i^{gt} \in \{0, 1\}$, serving as a simulation or "proxy" of ground truth. While many variations exist, the goodness of $\hat{\boldsymbol{e}}$ can be defined by, e.g., their intersection divided by their union.

$$\Psi_{\text{L}}(\hat{\boldsymbol{e}}, \boldsymbol{s}^{gt}) = \frac{\hat{\boldsymbol{e}} \cap \boldsymbol{s}^{gt}}{\hat{\boldsymbol{e}} \cup \boldsymbol{s}^{gt}}. \tag{15}$$

**Axiomatic** ([Sundararajan et al., 2017](#); [Nguyen & Martinez, 2020](#)) estimators measure to what extent an explanation fulfil some axiomatic properties such as completeness ([Sundararajan et al., 2017](#)) and non—sensitivity ([Nguyen & Martinez, 2020](#)). Due to the ambiguity that arises when empirically evaluating estimators in this category, we do not study this category in detail.

### A.3 Explanation Quality: Estimator Definitions

Within each of the five categories of explanation quality used in this work; (a) faithfulness, (b) robustness, (c) randomisation, (d) complexity and (e) localisation, we benchmarked two estimators per category in our benchmarking experiments. In the experiments that involved comparing meta-evaluation outcomes of estimators within the localisation category across different model architectures, we included two additional estimators.

**Faithfulness Correlation (FC)** ([Bhatt et al., 2020](#)) is defined in the following:

$$\Psi_{\text{FC}} = \operatorname*{corr}_{S \in |S| \subseteq d} \left( \sum_{i \in S} \Phi(\boldsymbol{x}, f, \hat{y}; \lambda)_i, f(\boldsymbol{x}) - f\left(\boldsymbol{x}_{[\boldsymbol{x}_s = \overline{\boldsymbol{x}}_s]}\right) \right), \tag{16}$$

where $|S| \subseteq D$ is a subset of indices of a sample $\boldsymbol{x}$, $\overline{\boldsymbol{x}}$ is the chosen baseline value and $\boldsymbol{x}_{[\boldsymbol{x}_s = \overline{\boldsymbol{x}}_s]}$ is, therefore, the masked input, with indices chosen randomly. Since $f$ denotes a trained model with parameters $\theta$, for brevity, we denote $f(\boldsymbol{x}; \theta)$ as $f(\boldsymbol{x})$ where possible. Higher values indicate that the explanation method's assignment of attribution is correlated with the behaviour of the model and hence is preferred.

**Pixel-Flipping (PF)** (Bach et al., 2015) returns a curve of prediction scores over an iterative set of pixel replacements, which are sorted in descending order by the highest relevant pixel in the explanation $\Phi(\boldsymbol{x}, f, \hat{y}; \lambda)$. To return one evaluation score per input sample, we calculate the area under the curve (AUC) as follows:

$$\Psi_{\text{PF}} = \sum_{i=1}^{n} (\hat{y}_i + \hat{y}_{i+1}) \cdot \frac{p_{i+1} - p_i}{2} \tag{17}$$

where $n$ is the number of discrete perturbation steps, $p_i$ and $p_{i+1}$ are the x-values for the $i^{th}$ and $(i+1)^{th}$ perturbation steps and $\hat{y}_i$ and $\hat{y}_{i+1}$ are the prediction values. For faithful explanations, a steep degradation of prediction scores is expected when attributions are iteratively replaced in descending order. Therefore, a lower value of AUC is indicative of better performance.

**Max-Sensitivity (MS)** (Yeh et al., 2019) measures the maximum sensitivity of an explanation using a Monte Carlo sampling-based approximation. It is defined as follows:

$$\Psi_{\text{MC}} = \max_{\boldsymbol{x}+\delta \in \mathcal{N}_\epsilon(\boldsymbol{x}) \leq \ \varepsilon} \left[ \frac{\|\Phi(\boldsymbol{x}, f, \hat{y}; \lambda) - \Phi(\boldsymbol{x} + \delta, f, \hat{y}; \lambda)\|}{\|\boldsymbol{x}\|} \right], \tag{18}$$

where $\varepsilon$ defines the radius of a discrete, finite-sample neighborhood around each input sample $\boldsymbol{x}$. This neighborhood, denoted as $\mathcal{N}_\epsilon(\boldsymbol{x})$, includes all samples in the set $X$ that are within a distance of $\varepsilon$ from $\boldsymbol{x}$. A lower MS score is indicative of more robustness.

**Local Lipschitz Estimate (LLE)** (Alvarez-Melis & Jaakkola, 2018) works similarly to the Max-Sensitivity (MS) method and estimates the Lipschitz constant of the explanation, which is a measure of how much the explanation changes with respect to the input under slight perturbation, defined as $\delta$. The LLE method is defined as follows:

$$\Psi_{\text{LLE}} = \max_{\boldsymbol{x}+\delta \in \mathcal{N}_\epsilon(\boldsymbol{x}) \leq \ \epsilon} \frac{\|\Phi(\boldsymbol{x}, f, \hat{y}; \lambda) - \Phi(\boldsymbol{x} + \delta, f, \hat{y}; \lambda)\|_2}{\|\boldsymbol{x} - (\boldsymbol{x} + \delta)\|_2}, \tag{19}$$

where lower values indicate less change with respect to the change in input, which is desirable.

**Model Parameter Randomisation Test (MPR)** (Adebayo et al., 2018) measures the correlation between an explanation from a randomly parameterised model $f(\boldsymbol{x}; \mathcal{P}_{\mathbb{F}}(\theta; v)) = \hat{f}$ and the original model $f$ for each separate layer $v$ of the network. To generate one quality estimate per sample, we calculate the average of the correlation coefficients over all the layers in the network, denoted $V$:

$$\Psi_{\text{MPR}} = \frac{1}{V} \sum_{v=1}^{V} \text{corr}(\Phi^v(\boldsymbol{x}, f, \hat{y}; \lambda), \Phi^v(\boldsymbol{x}, \hat{f}, \hat{y}; \lambda)), \tag{20}$$

where $\Phi^v(\boldsymbol{x}, f, \hat{y}; \lambda)$ is the explanation generated by the original model $f$ for layer $v$ and $\Phi^v(\boldsymbol{x}, \hat{f}, \hat{y}; \lambda)$ is the explanation generated by the randomly parameterised model $\hat{f}$ for layer $v$. The correlation between the two explanations is calculated for each layer and then averaged over all layers to generate the MPR score, where a lower correlation coefficient is desired.

**Random Logit (RL)** method proposed by (Sixt et al., 2020) is originally defined using the structural similarity index ($SSIM$) over the explanation of the ground truth label and an explanation of non-target class $y'$. However, to make it comparable with the MPR estimator, the $SSIM$ calculation is replaced with the *Spearman Rank Correlation Coefficient* as follows:

$$\Psi_{\text{RL}} = \text{corr}(\Phi(\boldsymbol{x}, f, \hat{y}; \lambda), \Phi(\boldsymbol{x}, f, y'; \lambda)), \tag{21}$$

where $\Phi(\boldsymbol{x}, f, \hat{y}; \lambda)$ is the explanation generated for the prediction $\hat{y}$ and $\Phi(\boldsymbol{x}, f, y'; \lambda)$ is the explanation generated for a non-target class $y'$. Lower values indicate that the explanations are not correlated which is desirable.

**Sparseness (SP)**   (Chalasani et al., 2020) is a method for evaluating the sparsity of explanations and is defined as the Gini index of the explanation. It is calculated by summing the product of the ranks of the input features and their attributions and dividing by the sum of the attribution as follows:

$$\Psi_{\text{SP}} = \frac{\sum_{i=1}^{D}(2i - D - 1) \cdot \hat{\boldsymbol{e}}_i}{D(D-1)\sum_{i=1}^{D}\hat{\boldsymbol{e}}_i}, \tag{22}$$

a higher sparseness score indicates lower complexity of the explanation $\hat{\boldsymbol{e}}$, which is desirable.

**Complexity (CO)**   (Bhatt et al., 2020) is defined using the Shannon entropy calculation which measures the amount of uncertainty or randomness in the explanation map. It is calculated by summing the product of the probabilities of the attributions and the logarithm of the probabilities of the attributions:

$$\Psi_{\text{CO}} = \mathbb{E}_i\left[-\ln\left(\mathbb{P}_\Phi\right)\right] = -\sum_{i=1}^{D}\mathbb{P}_\Phi(i)\ln\left(\mathbb{P}_\Phi(i)\right)$$

$$\text{with} \quad \mathbb{P}_\Phi(i) = \frac{|\Phi_i(\boldsymbol{x}, f, \hat{y}; \lambda)|}{\sum_{j \in [d]}|\Phi_j(\boldsymbol{x}, f, \hat{y}; \lambda)|}; \mathbb{P}_\Phi = \{\mathbb{P}_\Phi(1), \ldots, \mathbb{P}_\Phi(d)\}, \tag{23}$$

where $|\cdot|$ denotes the absolute value, $\mathbb{P}_\Phi(i)$ denotes the fractional contribution of feature $\boldsymbol{x}_i$ to the total quantity of the attribution. A higher entropy indicates a higher level of uncertainty or randomness, i.e., a higher complexity. A uniformly distributed attribution would have the highest possible complexity score.

**Pointing-Game (PG)**   (Zhang et al., 2018) captures whether the feature of maximal attribution lies on the ground truth mask, which is a binary mask indicating the true features that contribute to the model's output. It is defined as follows:

$$\Psi_{\text{PG}} = \begin{cases} 1 & \text{if } \arg\max_i \Phi_i(\boldsymbol{x}, f, \hat{y}; \lambda) \in \boldsymbol{s}^{gt} \\ 0 & \text{otherwise} \end{cases} \tag{24}$$

where $\Phi_i(\boldsymbol{x}, f, \hat{y}; \lambda)$ represents the $i^{th}$ input feature of highest atttribution and $\boldsymbol{s}^{gt} \in \mathbb{R}^D$ denotes the binary ground truth mask.

**Relevance Mass Accuracy (RMA)**   (Arras et al., 2022) quantifies the fraction of the sum of the attribution that intersects with the ground truth mask over the full explanation sum. It is defined as follows:

$$\Psi_{\text{RMA}} = \frac{\sum_{i=1}^{D}\Phi_i(\boldsymbol{x}, f, \hat{y}; \lambda) \cdot \boldsymbol{s}_{gt,i}}{\sum_{i=1}^{D}\Phi_i(\boldsymbol{x}, f, \hat{y}; \lambda)}, \tag{25}$$

where $\Phi_i(\boldsymbol{x}, f, \hat{y}; \lambda)$ is the attribution of the $i^{th}$ input feature and $\boldsymbol{s}_{gt,i}$ is the value of the $i^{th}$ element in the ground truth mask.

**Top-K Intersection (TK)**   (Theiner et al., 2022) extends the *Pointing-Game* method (Zhang et al., 2018) by measuring the fraction of $K$ highest ranked attributions that lie on the binary mask $\boldsymbol{s}_{gt}$:

$$\Psi_{\text{TK}} = \frac{|\boldsymbol{s}_{1-K}|}{|K|} \quad \text{with} \quad \boldsymbol{s}_{1-K} = \boldsymbol{r}_{1-K} \cap \boldsymbol{s}_{gt}, \tag{26}$$

where $\boldsymbol{s}_K$ denotes the subset of indices of explanation $\hat{\boldsymbol{e}}$ that corresponds to the $K$ highest ranked features that intersect with $\boldsymbol{s}_{gt}$, with $\boldsymbol{r} = Rank(\hat{\boldsymbol{e}})$.

**Relevance Rank Accuracy (RRA)**   (Arras et al., 2022) counts the number of features ranked by attribution value that intersects with $\boldsymbol{s}_{gt}$:

$$\Psi_{\text{RRA}} = \frac{|\boldsymbol{s}_{\hat{e}}|}{|\boldsymbol{s}_{gt}|} \quad \text{with} \quad \boldsymbol{s}_{\hat{e}} = \boldsymbol{r} \cap \boldsymbol{s}_{gt}, \tag{27}$$

where $\boldsymbol{s}_{\hat{e}}$ represents the elements of the explanation $\hat{\boldsymbol{e}}$ that intersect with each $i^{th}$ positive element in the ground truth mask $\boldsymbol{s}_{gt}$, with $\boldsymbol{r} = Rank(\hat{\boldsymbol{e}})$.

## A.4   Experimental Setup

In this section, we describe the experimental setup more in detail, which includes the datasets, models, explanation methods and estimators used in this work. We keep this section short since most of the methods in the following have been widely discussed in previous works. For more details, we refer the reader to the respective original publications. The experiments can be reproduced following the instructions in the repository (https://github.com/annahedstroem/MetaQuantus).

**Datasets**   We use four image classification datasets in the experiments: ILSVRC-15 (ImageNet) (Russakovsky et al., 2015), MNIST(LeCun et al., 2010), fMNIST (Xiao et al., 2017) and cMINST (customised-MNIST) (Bykov et al., 2022). For MNIST and fMNIST, we randomly sample 1024 test samples. We randomly sample 384 test samples from cMINST (customised-MNIST) (Bykov et al., 2022) which consists of MNIST digits displayed on uniformly sampled CIFAR-10 (Krizhevsky et al., 2009) backgrounds. To understand the real impact of State-of-the-art (SOTA), we also perform experiments on ILSVRC-15 (ImageNet) (Russakovsky et al., 2015), using 206 randomly selected test samples in batches of 50 samples.

We have chosen these datasets based on the availability of segmentation masks, since the estimators within the localisation category require such masks for computation. The bounding boxes for these datasets are designed to enclose the object of interest. For cMNIST, the bounding box covers 25% of the input and for MNIST and fMNIST, they cover approximately 35% (but up to 64%) of the input features. For ImageNet, the bounding boxes vary in size depending on the class of interest.

**Models**   The experiments are performed using different neural network models, including architectures such as LeNets (LeCun et al., 1998) and ResNets (He et al., 2016) which contributes to the robustness of our findings. For MNIST and fMNIST, we train LeNets to an accuracy of 98.14% and 87.44% respectively. For the cMNIST dataset, a ResNet-9 is trained to an accuracy of 98.09%. The training of all models is performed in a similar fashion; employing SGD optimisation with a standard cross-entropy loss, an initial learning rate of 0.001 and momentum of 0.9. All models are trained for 20 epochs each. For ILSVRC-15 (Russakovsky et al., 2015), we use the ResNet-18 model (He et al., 2016) with pre-trained weights given the ImageNet dataset, accessible via PyTorch (Paszke et al., 2019). In Appendix A.6.4, we further performed experiments with transformers-based model architectures including Vision Transformer (ViT) (Dosovitskiy et al., 2021) and SWIN Transformer (Liu et al., 2021b), both with pre-trained weights using PyTorch (Paszke et al., 2019).

**Explanation methods**   We employ explanation methods from a widely used category of post-hoc attribution methods, both gradient-based and model-agnostic techniques, i.e., *Gradient* (Morch et al., 1995; Baehrens et al., 2010), *Saliency* (Morch et al., 1995), *GradCAM* (Selvaraju et al., 2020), *Integrated Gradients* (Sundararajan et al., 2017), *Input×Gradient* (Shrikumar et al., 2016), *Occlusion* (Zeiler & Fergus, 2014) and *GradientSHAP* (Lundberg & Lee, 2017).

In all experiments, we generate explanations for a sample's predicted class $\hat{y}$. Whereas certain estimators such as the Saliency explanation ignore the signs of the explanations, we refrain from taking their absolute values, to preserve the explainable evidence in the attribution. For comparability, we normalise the explanations prior to their evaluation using the square root of its average second-moment estimate (Binder et al., 2022), which is defined as follows:

$$\frac{\hat{e}_{h,w}}{\left(\frac{1}{HW} \sum_{h',w'} \hat{e}_{h',w'}^2\right)^{1/2}} \, , \tag{28}$$

where $\hat{e}_{h,w}$ is the value of the explanation map at pixel location $(h, w)$ and $H$, $W$ denote the height and width, respectively[7].

---

[7]This normalisation ensures that each score in the attribution map has an average squared distance to zero that is equal to one. Since this operation does not normalise the attributions into a fixed range, it is not meant for visualisations, rather it is meant to preserve a quantity that is useful for the comparison of distances between different explanation methods.

**Estimators** We select the most established estimators within each of the five categories of explanation quality: *Complexity* (CO) (Bhatt et al., 2020), *Sparseness* (SP) (Chalasani et al., 2020), *Faithfulness Correlation* (FC) (Bhatt et al., 2020), *Pixel-Flipping* (PF) (Bach et al., 2015), *Max-Sensitivity* (MS) (Yeh et al., 2019), *Local Lipschitz Estimate* (LLE) (Alvarez-Melis & Jaakkola, 2018), *Pointing-Game* (PG) (Zhang et al., 2018), *Relevance Mass Accuracy* (RMA) (Arras et al., 2022), *Model Parameter Randomisation Test* (MPR) (Adebayo et al., 2018) and *Random Logit* (RL) (Sixt et al., 2020). We have defined each of the individual estimators mathematically in Appendix A.3.

**Parameterisation** For the initialisation of the different estimators, we mostly followed the recommendations as stated in the respective original publications. However, to make the estimators within a certain explanation category as comparable as possible, alterations to certain hyperparameters were made. When applying *Pixel-Flipping* (Bach et al., 2015) on image datasets, it generally becomes computationally infeasible to iterate over one pixel at a time. Therefore, we iterate over $\frac{2*w}{D}$ where $D$ is the dimensions of the input and $w$ and $h$ are the width and height of the image, respectively (which are assumed to be the same). We also use this same value to choose the subset size for *Faithfulness Correlation* (Bhatt et al., 2020). For both faithfulness estimators, as the replacement strategy for masked pixels, we use uniform sampling where we set the lower and higher bounds to the minimum and the maximum value of the test set, respectively. For the robustness estimators, which both are based on Monte Carlo sampling-based approximation, we let it run for 10 iterations. In the randomisation category, for comparability, we use the *Spearman's Rank Correlation Coefficient* to calculate the similarity between the original explanation and the explanation subject to randomisation. A full overview of the parameterisation of the estimators can be found in the GitHub repository `https://github.com/annahedstroem/MetaQuantus`.

**Hardware** All experiments were computed on GPUs where we used NVIDIA A100-PCIE 40GB for the toy datasets and NVIDIA A100-PCIE 80GB and Tesla V100S-PCIE-32GB for ImageNet dataset.

## A.5 Sanity-Checks of the Meta-Evaluation Framework

In this section, we conduct two sanity-checking experiments. In the first experiment, we create and meta-evaluate adversarial estimators to demonstrate the usability of the framework in practice and highlight how the two failure modes act complementary. In the second experiment, we examine the extent that the choice of $L$, i.e., the set of explanation methods, may influence the MC score.

### A.5.1 Adversarial Estimators

To sanity-check the meta-evaluation framework, we created adversarial quality estimators that were intended to perform poorly in a specific failure mode and thus, should indisputably fail the corresponding part of the testing scenarios of IPT and MPT. Specifically, we created an adversarial quality estimator that, independent of its given model, data and explanations, returns scores that are always the same (i.e., using deterministic sampling[8]). As such, this estimator should ultimately fail the reactivity to adversary tests (i.e., IAC$_{AR}$ and IEC$_{AR}$) since those tests expect a response to disruption. We denote this estimator $\Psi_=$. We create a second adversarial quality estimator that, independent of its inputs, returns scores that are drawn from a different probability distribution (i.e., using stochastic sampling[9]). In this setup, we expect poor performance on the noise resilience tests (i.e., IAC$_{NR}$ and IEC$_{NR}$) since these tests check that the quality estimates remain relatively unchanged after perturbation. We denote this adversarial estimator $\Psi_{\neq}$.

Table 2 summarises the outcome of this exercise, which includes the four score elements IAC$_{NR}$, IAC$_{AR}$, IEC$_{NR}$ and IEC$_{AR}$, aggregated for 5 iterations with $K = 10$ for the two tests, IPT and MPT. The expectation of the test outcome is indicated by the value in brackets after the display of the actual score, including the

---

[8]We assemble this adversarial estimator by repeatedly returning the same scores for $q'$ as one set of uniformly sampled scores $\hat{q} \sim \mathcal{U}(\alpha, \beta)$ with $\alpha = 0$ and $\beta = 1$.

[9]Here, we sample from a normal distribution $\mathcal{N}(\mu, \sigma^2)$ with $\sigma^2 = 1$ but with different means for the unperturbed- and the perturbed estimates, respectively. For the unperturbed estimates $\hat{q}$, we sample $\mu$ from a wide range, i.e., $\mu \in [-100000, -1]$ and for the perturbed estimates $q'$, we set a narrow range with $\mu \in [0, 1]$.

standard deviation. Here, a value of 0 indicates that the test should fail[10] and any other value indicates the desired outcome of the test to be successful. From Table 2, we note that both estimators, $\Psi_{\neq}$ and $\Psi_{=}$ produced scores that align with the expected value. Since estimator $\Psi_{\neq}$, relies on stochastic sampling, the scores are approximate, nevertheless, the scores and expectation are close and the standard deviation is small, indicating that the sanity checks results are stable. Overall, we can observe that the expectation aligns with the empirical reality across the different test settings. Therefore, we conclude the sanity-checking experiment to be passed.

Another important insight that can be drawn from Table 2 is that the two failure modes complement each other in determining the performance of an estimator. For an estimator that is provably bad, i.e., returns scores that are completely unrelated to the model, data and explanation methods (such as demonstrated by $\Psi_{\neq}$ and $\Psi_{=}$), at least one of the failure modes (AR or NR) will reveal that the estimator is failing. To fully assess the performance of an estimator, both failure modes are therefore necessary.

Table 2: The sanity-check exercise results show aggregated scores including std, over 5 iterations with $K = 5$. The direction of the arrow, i.e., $\uparrow$ indicates if a higher value is better. The expectation of the test outcome is indicated by the value in brackets, after the display of the actual score.

| Test | Estimator | $\mathbf{IAC}_{NR}$ ($\uparrow$) | $\mathbf{IAC}_{AR}$ ($\uparrow$) | $\mathbf{IEC}_{NR}$ ($\uparrow$) | $\mathbf{IEC}_{AR}$ ($\uparrow$) |
|------|-----------|------------------|------------------|------------------|------------------|
| IPT | $\Psi_{\neq}$ | $0.015 \pm 0.023$ (0.00) | $0.983 \pm 0.011$ (1.00) | $0.248 \pm 0.004$ (0.25) | $0.0 \pm 0.0$ (0.00) |
| | $\Psi_{=}$ | $1.0 \pm 0.0$ (1.00) | $0.0 \pm 0.0$ (0.00) | $1.0 \pm 0.0$ (1.00) | $0.0 \pm 0.0$ (0.00) |
| MPT | $\Psi_{\neq}$ | $0.014 \pm 0.010$ (0.00) | $0.973 \pm 0.019$ (1.00) | $0.248 \pm 0.003$ (0.25) | $0.0 \pm 0.0$ (0.00) |
| | $\Psi_{=}$ | $1.0 \pm 0.0$ (1.00) | $0.0 \pm 0.0$ (0.00) | $1.0 \pm 0.0$ (1.00) | $0.0 \pm 0.0$ (0.00) |

### A.5.2 Dependency on L

The meta-evaluation framework is intentionally designed to take into account the set of explanation methods given in the setup. For example, in the inter-consistency criterion (IEC) for noise resilience, we compute the estimator's ability to rank different explanation methods consistently when exposed to minor perturbations. The resulting MC score of a quality estimator will, therefore, to a certain extent, be dependent on the choice of $L$: both in terms of its cardinality and how similar the explanation functions are.

To understand how the performance of our quality estimator may vary depending on the choice of $L$, we conducted an experiment where we computed the MC score while enumerating various choices of $L$. In this experiment, we vary both the cardinality of $L$, by choosing values of $\{2, 3, 4\}$ and the methods included in the set. We selected both model-agnostic explanation methods such as *Occlusion* (Zeiler & Fergus, 2014) as well as gradient-based techniques such as *GradCAM* (Selvaraju et al., 2020), *Integrated Gradients* (Sundararajan et al., 2017) etc. For the sets of 2 explanation methods we included: $\{Gradient, Occlusion\}$, $\{Gradient, Input\times Gradient\}$, $\{Gradient, Saliency\}$, $\{Gradient, Input\times Gradient\}$. For sets with 3 methods: $\{Gradient,$

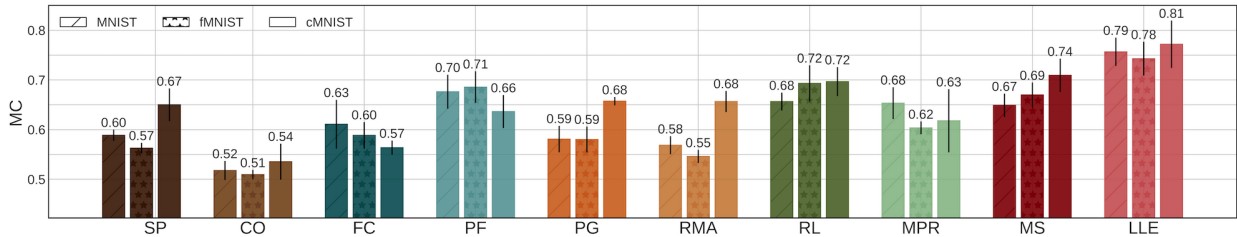

Figure 9: Comparison of averaged meta-consistency performance for different quality estimators using MPT and IPT, aggregated over 3 iterations with $K = 5$, across models $\{LeNet, ResNet\}$ and different datasets $\{MNIST, fMNIST, cMNIST\}$ with error bars showing the standard deviation.

---

[10]The exception is the expected value of the inter-consistency score, $\text{IEC}_{NR}$, for estimator $\Psi_{\neq}$ is not 0.0 but 0.25. This is because, for an estimator that assigns attributions randomly, i.e., independent of the explanation method, the likelihood of the condition $\bar{r}_j^M = \bar{r}_j^\star$ is $\frac{1}{L}$.

*GradCAM, GradientSHAP*}, {*Gradient, Saliency, Integrated Gradients*} and for 4 methods: {*Gradient, Saliency, Input×Gradient, GradCAM*}, {*Gradient, Saliency, Occlusion, GradCAM*}.

In Figure 9, we show the aggregate values for different explanation sets across the datasets separately. Here, the error bars indicate the standard deviations. By comparing the MC scores category by category, we can observe that the error bars from the respective estimator do generally not overlap. This means that the choice of $L$ has limited influence on the MC score, suggesting the measure's stability.

### A.6 Supplementary Experiments

In the following section, we present additional experiments conducted in the scope of this work. First, we demonstrate that `MetaQuantus` can be used for additional applications in Explainable AI. Here, we include two demonstrations, first, we show how the MC score can be employed as a target variable for optimising the hyperparameters of an estimator and second, we demonstrate how the framework can be used to analyse category convergence. At the end of this section, we discuss supplementary results for the benchmarking experiment which includes an additional analysis of ranking consistency.

#### A.6.1 Application — Hyperparameter Optimisation

It is practically well-known and increasingly publicly recognised (Bansal et al., 2020; Brunke et al., 2020; Brocki & Chung, 2022; Pahde et al., 2022) how difficult it can be to tune the hyperparameters in the explainability domain. Unlike in traditional ML, in XAI, we generally do not have a target variable to optimise against. As an additional experiment, we, therefore, investigated how the meta-evaluation framework can be useful in solving the task of selecting the best set of hyperparameters for a given estimator. For this, we choose an estimator with relatively many parameters, that is *Faithfulness Correlation* (Bhatt et al., 2020) and performed a grid-search on these using ImageNet. By exploring combinations of three baseline strategies = ['Black', 'Uniform', 'Mean'] and four subset sizes = [28, 52, 102, 128], we created 12 hyperparameter settings[11]. We determined the performance of each of the estimator's parameterisation by storing the meta-evaluation vector $m$ and the MC score at each run. The objective of this exercise is to determine the hyperparameter setting that optimises the performance of the estimator, i.e., its resilience to noise and reactivity to adversary.

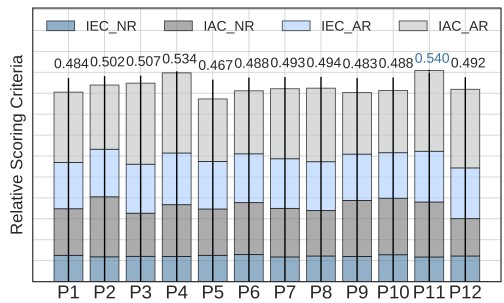
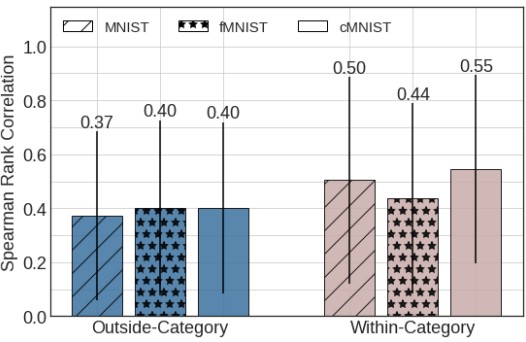

Figure 10: *Left:* The results of using the meta-evaluation framework to optimize the hyperparameters of FC (Bhatt et al., 2020) estimator across 12 parameterisations (P1-P12) on ImageNet dataset, averaged over 3 iterations with $K = 3$. The parameter setting P11 demonstrated the highest scores with small standard deviation and thus is selected as the parameter setting. *Right:* The results from comparing the correlation coefficients between the meta-evaluation vector scores for estimators within the same category versus those outside of the category, suggesting that the estimators of the same category have more resemble with respect to its performance characteristics compared to estimators outside.

---

[11]The parameters were combined in the following 12 settings: P1: ['Black', 28], P2: ['Black', 52], P3: ['Black', 102], P4: ['Black', 128], P5: ['Uniform', 28], P6: ['Uniform', 52], P7: ['Uniform', 102], P8: ['Uniform', 128], P9: ['Mean', 28], P10: ['Mean', 52], P11: ['Mean', 102], P12: ['Mean', 128].

From Figure 10 (left), we can observe that P11 has the highest meta-consistency score and as such, we recommend the associated parameter setting of "mean" as the replacement strategy with 102 features as the subset size. In contrast to previous works that found a relatively large difference in evaluation outcomes between different parameterisations of faithfulness estimators (Tomsett et al., 2020; Brunke et al., 2020; Rong et al., 2022; Hedström et al., 2023), we detect, that with the MC score—which provides a more comprehensive picture of the estimator's performance—there is not a considerable variability, as depicted by the similarity in IAC and IEC scores over P1 to P12.

### A.6.2 Application — Category Convergence

The question of whether quality estimators within the same category are measuring the same underlying concept has been of significant interest to the community (Tomsett et al., 2020; Gevaert et al., 2022). Based on the observed similarity of estimator shapes in Figure 5—that the estimators within the same category typically have a higher resemblance in area shapes compared to estimators outside of their categories—we sought to employ the meta-evaluation framework to investigate whether estimators within a category exhibit a greater level of correlation than those outside of the category. To address this question, often referred to as "convergent validity", the prevalent technique has been to measure intra-correlation, which simply involves correlating the scores of different estimators within the same category. This approach, however, has limitations, as it disregards the aspect of ranking consistency (IEC) and may not account for the fact that scores from different estimators may have different scales and interpretations, which may skew the results.

We improve upon the current methodology proposed in Tomsett et al. (2020); Gevaert et al. (2022) by calculating the correlation coefficient on the meta-evaluation vector $\boldsymbol{m}$ of different estimators, within- and outside of their category as produced in the benchmarking exercise. This approach is advantageous as it: (i) yields scores in a normalised range $[0, 1]$ and (ii) provides a more comprehensive view of the estimator's performance characteristics by incorporating multiple failure modes and criteria.

Figure 10 (right) presents the results of this experiment, averaged over all estimators as described in A.4. Here, we can observe that the estimator's performance characteristics are more similar within a category, as seen in the higher correlation coefficient (*Spearman Rank Correlation Coefficient*) across all datasets. These observations contrast previous works by Tomsett et al. (2020); Gevaert et al. (2022) that found a low correlation coefficient (for faithfulness estimators in particular). We posit that this difference can be explained by the fact that the meta-evaluation framework considers multiple failure modes and criteria of

Table 3: Meta-evaluation benchmarking results with MNIST, aggregated over 3 iterations with $K = 5$. IPT results are in grey rows and MPT results are in white rows. $\overline{\text{MC}}$ denotes the averages of the MC scores over IPT and MPT. The top-performing MC- or $\overline{\text{MC}}$ method in each explanation category, which outperforms the bottom-performing method by at least 2 standard deviations, is underlined. Higher values are preferred for all scoring criteria.

| Category | Estimator | $\overline{\text{MC}}$ ($\uparrow$) | MC ($\uparrow$) | IAC$_{NR}$ ($\uparrow$) | IAC$_{AR}$ ($\uparrow$) | IEC$_{NR}$ ($\uparrow$) | IEC$_{AR}$ ($\uparrow$) |
|---|---|---|---|---|---|---|---|
| *Complexity* | Sparseness | $\underline{0.558 \pm 0.028}$ | $0.640 \pm 0.043$ | $0.209 \pm 0.040$ | $0.946 \pm 0.086$ | $0.837 \pm 0.002$ | $0.569 \pm 0.046$ |
| | | | $0.476 \pm 0.013$ | $0.929 \pm 0.063$ | $0.053 \pm 0.014$ | $0.840 \pm 0.005$ | $0.084 \pm 0.001$ |
| | Complexity | $0.521 \pm 0.003$ | $0.541 \pm 0.007$ | $0.009 \pm 0.013$ | $1.000 \pm 0.000$ | $1.000 \pm 0.000$ | $0.156 \pm 0.014$ |
| | | | $0.500 \pm 0.000$ | $0.167 \pm 0.000$ | $0.833 \pm 0.000$ | $1.000 \pm 0.000$ | $0.000 \pm 0.000$ |
| *Faithfulness* | Faithfulness Corr. | $0.540 \pm 0.015$ | $0.537 \pm 0.003$ | $0.477 \pm 0.032$ | $0.900 \pm 0.023$ | $0.190 \pm 0.003$ | $0.579 \pm 0.008$ |
| | | | $0.543 \pm 0.026$ | $0.500 \pm 0.107$ | $0.890 \pm 0.005$ | $0.190 \pm 0.002$ | $0.594 \pm 0.005$ |
| | Pixel-Flipping | $\underline{0.626 \pm 0.039}$ | $0.609 \pm 0.039$ | $0.547 \pm 0.139$ | $0.963 \pm 0.034$ | $0.299 \pm 0.001$ | $0.626 \pm 0.046$ |
| | | | $0.644 \pm 0.038$ | $0.485 \pm 0.141$ | $1.000 \pm 0.000$ | $0.294 \pm 0.006$ | $0.796 \pm 0.006$ |
| *Localisation* | Pointing-Game | $\underline{0.586 \pm 0.010}$ | $0.672 \pm 0.020$ | $0.977 \pm 0.005$ | $0.607 \pm 0.075$ | $0.996 \pm 0.000$ | $0.108 \pm 0.012$ |
| | | | $0.500 \pm 0.000$ | $1.000 \pm 0.000$ | $0.000 \pm 0.000$ | $1.000 \pm 0.000$ | $0.000 \pm 0.000$ |
| | Relevance Mass Acc. | $0.552 \pm 0.015$ | $0.613 \pm 0.022$ | $0.258 \pm 0.062$ | $0.793 \pm 0.023$ | $0.846 \pm 0.001$ | $0.553 \pm 0.032$ |
| | | | $0.491 \pm 0.007$ | $0.940 \pm 0.019$ | $0.071 \pm 0.019$ | $0.902 \pm 0.003$ | $0.051 \pm 0.000$ |
| *Randomisation* | Random Logit | $\underline{0.666 \pm 0.004}$ | $0.712 \pm 0.008$ | $0.360 \pm 0.041$ | $0.969 \pm 0.010$ | $0.937 \pm 0.003$ | $0.581 \pm 0.006$ |
| | | | $0.620 \pm 0.000$ | $0.186 \pm 0.000$ | $0.874 \pm 0.000$ | $0.860 \pm 0.000$ | $0.562 \pm 0.000$ |
| | Model Param. Rand. | $0.583 \pm 0.007$ | $0.624 \pm 0.005$ | $0.264 \pm 0.019$ | $0.959 \pm 0.000$ | $0.764 \pm 0.002$ | $0.510 \pm 0.001$ |
| | | | $0.542 \pm 0.010$ | $0.250 \pm 0.065$ | $0.806 \pm 0.028$ | $0.647 \pm 0.003$ | $0.463 \pm 0.004$ |
| *Robustness* | Max-Sensitivity | $0.649 \pm 0.007$ | $0.754 \pm 0.002$ | $0.547 \pm 0.064$ | $0.938 \pm 0.033$ | $0.804 \pm 0.001$ | $0.726 \pm 0.038$ |
| | | | $0.545 \pm 0.012$ | $0.361 \pm 0.053$ | $1.000 \pm 0.000$ | $0.806 \pm 0.005$ | $0.011 \pm 0.001$ |
| | Local Lipschitz Est. | $\underline{0.741 \pm 0.030}$ | $0.726 \pm 0.026$ | $0.484 \pm 0.091$ | $0.935 \pm 0.088$ | $0.736 \pm 0.002$ | $0.750 \pm 0.034$ |
| | | | $0.756 \pm 0.034$ | $0.519 \pm 0.118$ | $0.974 \pm 0.017$ | $0.740 \pm 0.005$ | $0.789 \pm 0.006$ |

Table 4: Meta-evaluation benchmarking results with fMNIST, aggregated over 3 iterations with $K = 5$. IPT results are in grey rows and MPT results are in white rows. $\overline{\text{MC}}$ denotes the averages of the MC scores over IPT and MPT. The top-performing MC- or $\overline{\text{MC}}$ method in each explanation category, which outperforms the bottom-performing method by at least 2 standard deviations, is underlined. Higher values are preferred for all scoring criteria.

| Category | Estimator | $\overline{\text{MC}}$ (↑) | MC (↑) | $\text{IAC}_{NR}$ (↑) | $\text{IAC}_{AR}$ (↑) | $\text{IEC}_{NR}$ (↑) | $\text{IEC}_{AR}$ (↑) |
|---|---|---|---|---|---|---|---|
| *Complexity* | Sparseness | $0.536 \pm 0.011$ | $0.596 \pm 0.012$ | $0.145 \pm 0.039$ | $0.915 \pm 0.045$ | $0.831 \pm 0.004$ | $0.492 \pm 0.082$ |
| | | | $0.475 \pm 0.010$ | $0.917 \pm 0.036$ | $0.070 \pm 0.003$ | $0.832 \pm 0.003$ | $0.083 \pm 0.001$ |
| | Complexity | $0.516 \pm 0.007$ | $0.532 \pm 0.014$ | $0.050 \pm 0.047$ | $0.990 \pm 0.027$ | $0.999 \pm 0.000$ | $0.086 \pm 0.028$ |
| | | | $0.500 \pm 0.000$ | $0.167 \pm 0.000$ | $0.833 \pm 0.000$ | $1.000 \pm 0.000$ | $0.000 \pm 0.000$ |
| *Faithfulness* | Faithfulness Corr. | $0.530 \pm 0.021$ | $0.524 \pm 0.021$ | $0.527 \pm 0.030$ | $0.857 \pm 0.072$ | $0.198 \pm 0.008$ | $0.515 \pm 0.004$ |
| | | | $0.536 \pm 0.021$ | $0.448 \pm 0.087$ | $0.994 \pm 0.003$ | $0.196 \pm 0.004$ | $0.504 \pm 0.002$ |
| | Pixel-Flipping | $0.530 \pm 0.021$ | $0.573 \pm 0.025$ | $0.447 \pm 0.050$ | $0.958 \pm 0.088$ | $0.329 \pm 0.002$ | $0.558 \pm 0.032$ |
| | | | $0.649 \pm 0.018$ | $0.453 \pm 0.073$ | $1.000 \pm 0.000$ | $0.324 \pm 0.001$ | $0.817 \pm 0.003$ |
| *Localisation* | Pointing-Game | $\underline{0.583 \pm 0.005}$ | $0.666 \pm 0.009$ | $0.950 \pm 0.025$ | $0.634 \pm 0.032$ | $0.995 \pm 0.001$ | $0.084 \pm 0.018$ |
| | | | $0.500 \pm 0.000$ | $1.000 \pm 0.000$ | $0.000 \pm 0.000$ | $1.000 \pm 0.000$ | $0.000 \pm 0.000$ |
| | Relevance Mass Acc. | $0.538 \pm 0.023$ | $0.587 \pm 0.024$ | $0.231 \pm 0.102$ | $0.806 \pm 0.056$ | $0.850 \pm 0.007$ | $0.460 \pm 0.048$ |
| | | | $0.490 \pm 0.022$ | $0.944 \pm 0.034$ | $0.067 \pm 0.067$ | $0.894 \pm 0.003$ | $0.055 \pm 0.003$ |
| *Randomisation* | Random Logit | $\underline{0.689 \pm 0.005}$ | $\underline{0.717 \pm 0.010}$ | $0.234 \pm 0.039$ | $1.000 \pm 0.000$ | $0.955 \pm 0.005$ | $0.680 \pm 0.005$ |
| | | | $0.660 \pm 0.000$ | $0.062 \pm 0.000$ | $1.000 \pm 0.000$ | $0.902 \pm 0.000$ | $0.677 \pm 0.000$ |
| | Model Param. Rand. | $0.570 \pm 0.010$ | $0.622 \pm 0.010$ | $0.355 \pm 0.042$ | $0.925 \pm 0.000$ | $0.755 \pm 0.005$ | $0.451 \pm 0.000$ |
| | | | $0.518 \pm 0.010$ | $0.098 \pm 0.008$ | $0.902 \pm 0.045$ | $0.657 \pm 0.004$ | $0.414 \pm 0.001$ |
| *Robustness* | Max-Sensitivity | $0.639 \pm 0.036$ | $0.699 \pm 0.037$ | $0.515 \pm 0.097$ | $0.961 \pm 0.021$ | $0.816 \pm 0.007$ | $0.501 \pm 0.058$ |
| | | | $0.580 \pm 0.035$ | $0.504 \pm 0.141$ | $1.000 \pm 0.000$ | $0.811 \pm 0.002$ | $0.004 \pm 0.000$ |
| | Local Lipschitz Est. | $\underline{0.710 \pm 0.022}$ | $0.696 \pm 0.038$ | $0.538 \pm 0.139$ | $0.979 \pm 0.033$ | $0.775 \pm 0.005$ | $0.492 \pm 0.092$ |
| | | | $\underline{0.724 \pm 0.006}$ | $0.567 \pm 0.037$ | $0.896 \pm 0.024$ | $0.774 \pm 0.001$ | $0.661 \pm 0.006$ |

Table 5: Meta-evaluation benchmarking results with cMNIST, aggregated over 3 iterations with $K = 5$. IPT results are in grey rows and MPT results are in white rows. $\overline{\text{MC}}$ denotes the averages of the MC scores over IPT and MPT. The top-performing MC- or $\overline{\text{MC}}$ method in each explanation category, which outperforms the bottom-performing method by at least 2 standard deviations, is underlined. Higher values are preferred for all scoring criteria.

| Category | Estimator | $\overline{\text{MC}}$ (↑) | MC (↑) | $\text{IAC}_{NR}$ (↑) | $\text{IAC}_{AR}$ (↑) | $\text{IEC}_{NR}$ (↑) | $\text{IEC}_{AR}$ (↑) |
|---|---|---|---|---|---|---|---|
| *Complexity* | Sparseness | $\underline{0.616 \pm 0.015}$ | $0.706 \pm 0.013$ | $0.352 \pm 0.061$ | $0.989 \pm 0.017$ | $0.814 \pm 0.001$ | $0.670 \pm 0.016$ |
| | | | $0.525 \pm 0.018$ | $0.626 \pm 0.099$ | $0.313 \pm 0.028$ | $0.830 \pm 0.005$ | $0.333 \pm 0.006$ |
| | Complexity | $0.541 \pm 0.018$ | $0.565 \pm 0.024$ | $0.056 \pm 0.084$ | $1.000 \pm 0.000$ | $0.996 \pm 0.001$ | $0.209 \pm 0.013$ |
| | | | $0.518 \pm 0.013$ | $0.062 \pm 0.010$ | $0.928 \pm 0.047$ | $1.000 \pm 0.000$ | $0.080 \pm 0.005$ |
| *Faithfulness* | Faithfulness Corr. | $0.562 \pm 0.014$ | $0.563 \pm 0.017$ | $0.508 \pm 0.061$ | $0.939 \pm 0.017$ | $0.182 \pm 0.004$ | $0.622 \pm 0.005$ |
| | | | $0.562 \pm 0.010$ | $0.490 \pm 0.031$ | $0.934 \pm 0.018$ | $0.188 \pm 0.008$ | $0.634 \pm 0.012$ |
| | Pixel-Flipping | $\underline{0.604 \pm 0.016}$ | $0.586 \pm 0.022$ | $0.565 \pm 0.040$ | $0.965 \pm 0.022$ | $0.287 \pm 0.005$ | $0.526 \pm 0.080$ |
| | | | $0.621 \pm 0.010$ | $0.495 \pm 0.037$ | $0.995 \pm 0.001$ | $0.295 \pm 0.012$ | $0.701 \pm 0.002$ |
| *Localisation* | Pointing-Game | $\underline{0.687 \pm 0.006}$ | $\underline{0.873 \pm 0.010}$ | $0.967 \pm 0.000$ | $1.000 \pm 0.000$ | $0.997 \pm 0.000$ | $0.527 \pm 0.040$ |
| | | | $0.502 \pm 0.001$ | $0.995 \pm 0.003$ | $0.013 \pm 0.003$ | $0.999 \pm 0.001$ | $0.001 \pm 0.000$ |
| | Relevance Mass Acc. | $0.621 \pm 0.011$ | $0.856 \pm 0.020$ | $0.751 \pm 0.008$ | $0.358 \pm 0.055$ | $1.000 \pm 0.000$ | $0.791 \pm 0.012$ |
| | | | $0.491 \pm 0.014$ | $0.640 \pm 0.028$ | $0.306 \pm 0.032$ | $0.796 \pm 0.003$ | $0.223 \pm 0.005$ |
| *Randomisation* | Random Logit | $\underline{0.713 \pm 0.005}$ | $0.723 \pm 0.010$ | $0.530 \pm 0.065$ | $0.894 \pm 0.026$ | $0.881 \pm 0.007$ | $0.586 \pm 0.012$ |
| | | | $0.703 \pm 0.000$ | $0.410 \pm 0.000$ | $0.884 \pm 0.000$ | $0.913 \pm 0.000$ | $0.606 \pm 0.000$ |
| | Model Param. Rand. | $0.657 \pm 0.009$ | $0.673 \pm 0.003$ | $0.490 \pm 0.006$ | $1.000 \pm 0.000$ | $0.814 \pm 0.005$ | $0.387 \pm 0.000$ |
| | | | $0.641 \pm 0.016$ | $0.417 \pm 0.058$ | $1.000 \pm 0.000$ | $0.804 \pm 0.005$ | $0.344 \pm 0.004$ |
| *Robustness* | Max-Sensitivity | $0.637 \pm 0.030$ | $0.690 \pm 0.035$ | $0.494 \pm 0.069$ | $0.972 \pm 0.045$ | $0.687 \pm 0.004$ | $0.606 \pm 0.050$ |
| | | | $0.583 \pm 0.024$ | $0.582 \pm 0.079$ | $0.992 \pm 0.002$ | $0.680 \pm 0.019$ | $0.080 \pm 0.002$ |
| | Local Lipschitz Est. | $\underline{0.697 \pm 0.020}$ | $0.689 \pm 0.026$ | $0.548 \pm 0.077$ | $0.971 \pm 0.049$ | $0.628 \pm 0.007$ | $0.609 \pm 0.042$ |
| | | | $0.706 \pm 0.014$ | $0.508 \pm 0.047$ | $0.999 \pm 0.000$ | $0.630 \pm 0.007$ | $0.685 \pm 0.005$ |

what a quality estimator should fulfil and not only one, e.g., ranking consistency (Rong et al., 2022) and as such, give a more comprehensive answer. However, from the error bars in Figure 10, we also observe, that the correlation coefficients are greatly varying within each group. Further research is thus necessary to fully understand the extent to which estimators of the same explanation quality category measure the same underlying concept.

### A.6.3 Supplementary Results — Benchmarking

Similar to Table 1, we present the results of the IPT and the MPT for the MNIST, fMNIST and cMNIST datasets in Tables 3-5, respectively. The grey rows indicate the results from the IPT and the white rows

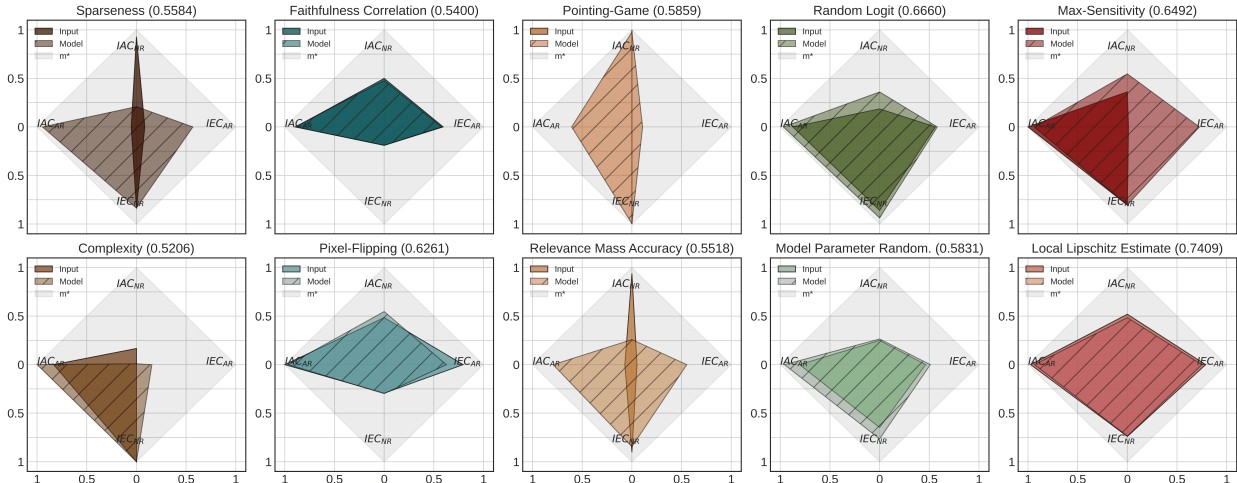

Figure 11: A graphical representation of the MNIST benchmarking results (Table 3), aggregated over 3 iterations with $K = 5$. Each column corresponds to a category of explanation quality, from left to right: *Complexity*, *Faithfulness*, *Localisation*, *Randomisation* and *Robustness*. The grey area indicates the area of an optimally performing estimator, i.e., $\mathbf{m}^* = \mathbb{1}^4$. The MC score (indicated in brackets) is averaged over MPT and IPT. Higher values are preferred.

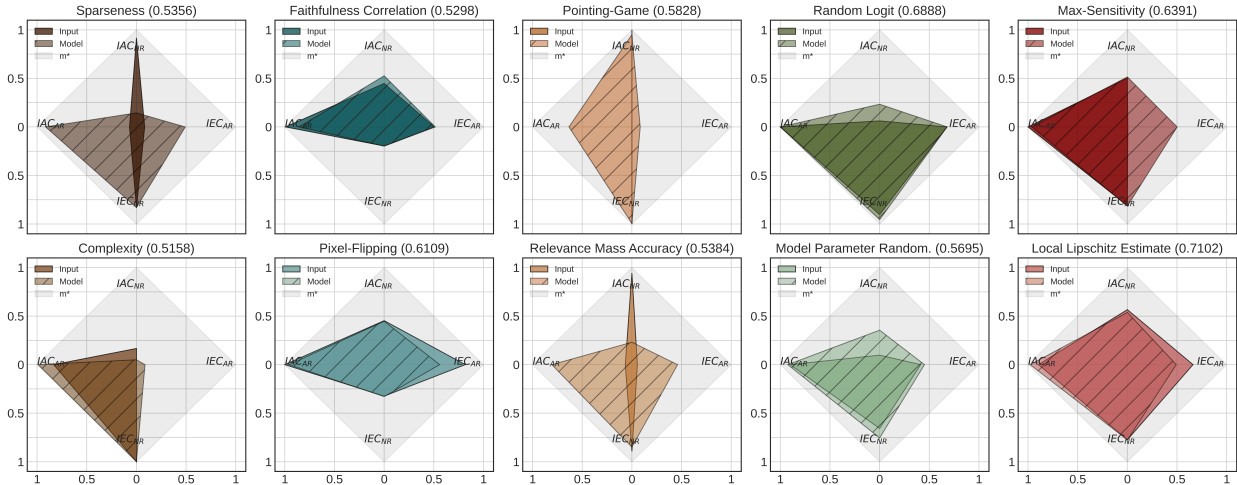

Figure 12: A graphical representation of the fMNIST benchmarking results (Table 4), aggregated over 3 iterations with $K = 5$. Each column corresponds to a category of explanation quality, from left to right: *Complexity*, *Faithfulness*, *Localisation*, *Randomisation* and *Robustness*. The grey area indicates the area of an optimally performing estimator, i.e., $\mathbf{m}^* = \mathbb{1}^4$. The MC score (indicated in brackets) is averaged over MPT and IPT. Higher values are preferred.

show the results from the MPT. The results are consistent with those presented in the main manuscript, both in terms of individual score criteria, estimator- and category comparison.

Similar to Figure 7, we also represent Tables 3-5 as area graphs. With an exception of slightly higher localisation scores for cMNIST dataset (as explained in the main paper), the results as demonstrated in Figures 11-13 are completely consistent with those findings presented in the main paper. Recall that, larger coloured areas imply better performance on the different scoring criteria and the grey area indicates the area of an optimally performing quality estimator, i.e., $\boldsymbol{m}^* = \mathbb{1}^4$. Each column of estimators represents a category of explanation quality, from left to right: *Complexity*, *Faithfulness*, *Localisation*, *Randomisation* and *Robustness*.

We further visualise the results (as shown in Tables 1, 4 and 5) as scatter plots for ImageNet, fMNIST and cMNIST datasets in Figure 14. In the main paper, we identified that the faithfulness category (blue points)

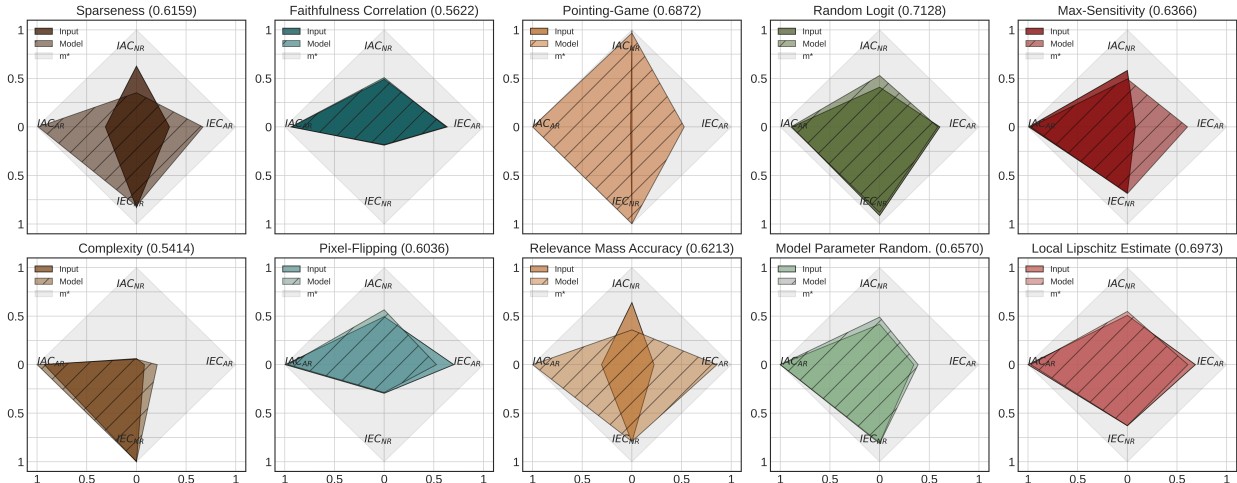

Figure 13: A graphical representation of the cMNIST benchmarking results (Table 5), aggregated over 3 iterations with $K = 5$. Each column corresponds to a category of explanation quality, from left to right: *Complexity*, *Faithfulness*, *Localisation*, *Randomisation* and *Robustness*. The grey area indicates the area of an optimally performing estimator, i.e., $\mathbf{m}^* = \mathbb{1}^4$. The MC score (indicated in brackets) is averaged over MPT and IPT. Higher values are preferred.

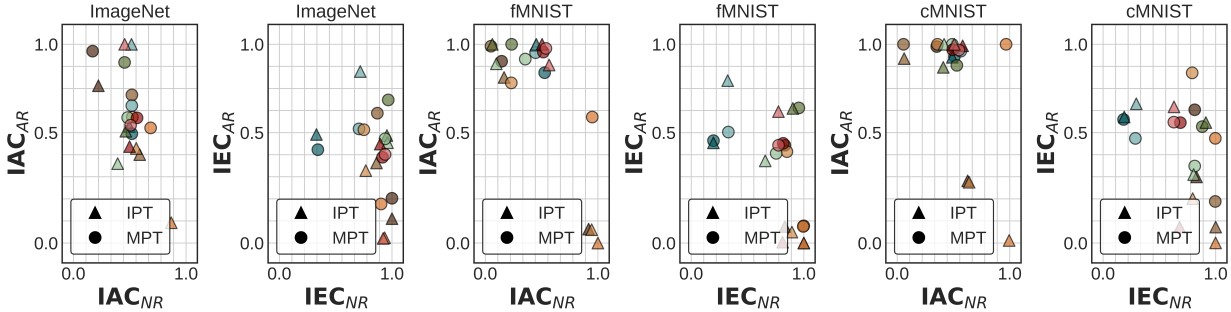

Figure 14: A supplementary visualisation of the benchmarking results (Tables 1, 4 and reftable-benchmarking-cmnist), in particular IAC and IEC scores for noise resilience (x-axes) and adverse reactivity (y-axes). The colours indicate the estimator and the symbols demonstrate the test: IPT and MPT, respectively. Higher values are preferred.

had a particularly low ranking consistency (IEC), which is also evident in these supplementary plots. From Figure 14, we can moreover observe how the estimators' scores on the respective failure modes are related. These plots also show that a higher resilience to noise does not necessarily imply more reactivity to adversary and vice versa—the performance characteristics of the estimators are more complex than that.

### A.6.4 Supplementary Results — Comparing Neural Network Architectures

As part of our exploration of meta-evaluation outcomes across varying model architectures, including those based on transformers, we embarked on additional benchmarking experiments. These involved comparing estimators from both the localisation and complexity categories of explanation quality among Vision Transformer (Dosovitskiy et al., 2021), SWIN (Liu et al., 2021b), and ResNet-18 (He et al., 2016) models using the ImageNet dataset (Russakovsky et al., 2015). The outcomes concerning the localisation category have been discussed in the main manuscript 6.1.3. The results for the complexity category, including estimators such as *Complexity* (CO) (Bhatt et al., 2020) and *Sparseness* (SP) (Chalasani et al., 2020) are discussed below. For this experiment we used two explanation methods $L = Input \times Gradient, Gradients$.

To analyse the results, we represent the entries of the meta-evaluation vector as coordinates on a 2D plane and visualise the benchmarking results as area graphs, as seen in Figure 15 (left). Again, larger coloured

areas imply better performance on the different scoring criteria and the grey area indicates the area of an optimally performing quality estimator, i.e., $\boldsymbol{m}^* = \mathbb{1}^4$. Each column represents the estimator and each row the network architecture, as indicated by the labels of Figure 15 (left).

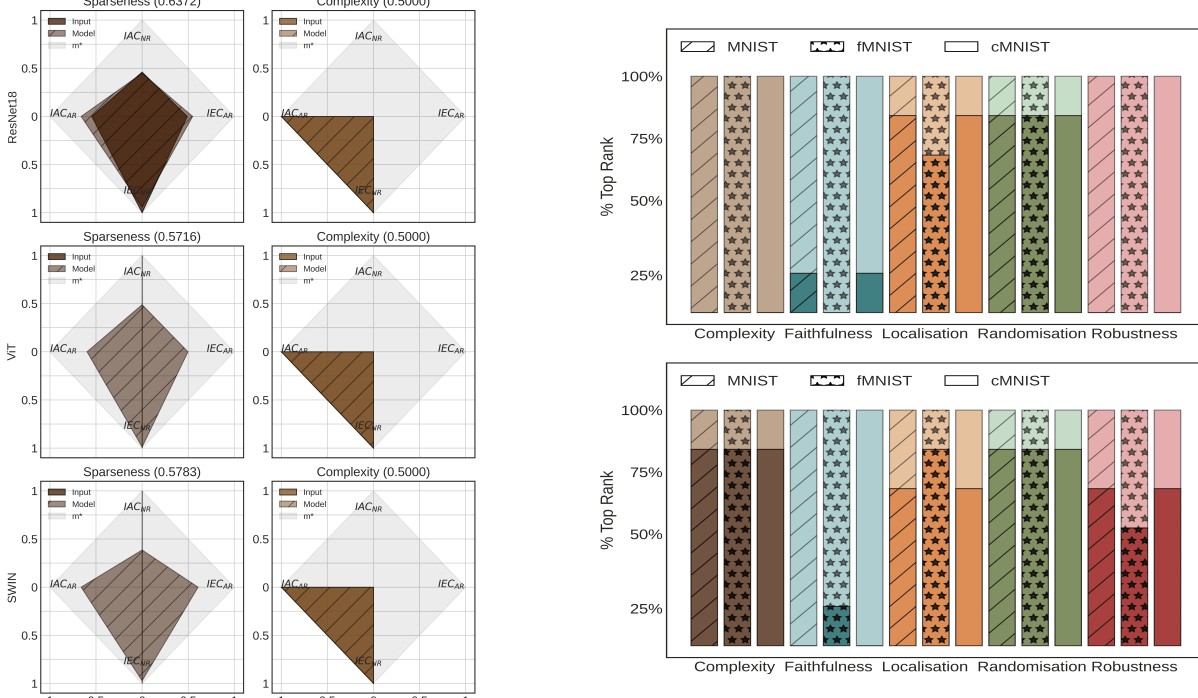

Figure 15: *Left:* A graphical representation of the ImageNet benchmarking results for the complexity category, aggregated over 3 iterations with $K = 5$. Each column represents the estimator, from left to right: *Sparseness* and *Complexity*, and each row the model type {*ResNet-18*, *ViT*, *SWIN*}, as indicated by the labels. The grey area indicates the area of an optimally performing estimator, i.e., $\mathbf{m}^* = \mathbb{1}^4$. The MC score (indicated in brackets) is averaged over MPT and IPT. Higher values are preferred. *Right:* A visualisation of the benchmarking results (Tables 3, 4 and 5) showing the distribution of top rankings within each category of explanation quality. There is higher variability in scores for the IPT tests (above), while it is lower for the MPT tests (below).

As can be observed in Figure 15 (left), the areas indicating the performance profiles of each individual estimator across the network types demonstrate a noticeable similarity. In accordance with the observations made in the main manuscript, the *Sparseness* method (Chalasani et al., 2020) demonstrates superior performance compared to the *Complexity* method (Bhatt et al., 2020). Overall, this provide additional evidence in proving the framework to be agnostic with respect to the choice of network architecture.

### A.6.5 Supplementary Results — Ranking Consistency

In the main paper, we presented the average MC scores for each dataset in Figure 5, which showed that the best-performing estimator in each category of explanation quality generally remained consistent across tested datasets. To further explore this consistency, we considered a margin of error of 2 standard deviations applied to the MC scores and re-calculated the within-category ranking for each individual estimator in each category and visualised the results in Figure 15 (right).

Figure 15 (right) showcases the the frequency with which the different estimators within each category achieved the highest or the lowest average score, respectively. The color scheme, consisting of one lighter and one darker hue for each category, is consistent with the individual estimators in previous figures. A larger fraction of a bar with a single hue indicates more "category wins" for a specific estimator. From Figure 15 (right), similar to observations in the main paper, we find that estimators within the localisation and randomisation categories are prone to stronger score variability. In general, we anticipate variations in

MC scores across all categories between different datasets. This is because all quality estimators evaluate explanation qualities, which depend on the specific dataset and network in use. Additionally, as seen in Figure 15 (right), the score variability for IPT (above) is higher than that of MPT (below). Therefore, in the main paper, we present the MC scores averaged over both MPT and IPT to capture a more comprehensive view of the estimator's performance.

### A.7 Notation Table

In the following, we provide notation tables that encompasses all the notations employed throughout this paper. The table is sorted according to the section in which each notation initially appeared.

#### Preliminaries

| | |
|---|---|
| $f$ | A black-box model function that maps input $\boldsymbol{x}$ to output $y$ |
| $\theta$ | The parameters of the model function $f$ |
| $\boldsymbol{X}_{\mathrm{tr}}$ | The training dataset on which the model $f$ is trained |
| $\boldsymbol{X}_{\mathrm{te}}$ | The test dataset on which the model $f$ is evaluated |
| $\boldsymbol{x}$ | An input in the instance space $\mathbb{X}$ |
| $y$ | An output class in the label space $\mathbb{Y}$ |
| $\hat{y}$ | A prediction made by the model $f$ |
| $C$ | The number of output classes |
| $N$ | The number of test samples |
| $D$ | The dimension of the input |
| $\mathbb{X}$ | The instance space |
| $\mathbb{Y}$ | The label space |
| $\mathbb{F}$ | The function space of all models |
| $\Phi$ | An explanation function that maps $\boldsymbol{x}$, $f$, and $\hat{y}$ to an explanation map $\hat{\boldsymbol{e}}$ |
| $\lambda$ | The parameter of the explanation function $\Phi$ |
| $\hat{\boldsymbol{e}}$ | The explanation map produced by $\Phi$ |
| $\mathbb{E}$ | The space of possible explanations |
| $\Psi$ | A quality estimation function that takes $\hat{\boldsymbol{e}}$ and returns a scalar $\hat{q}$ to indicate its quality |
| $\tau$ | The parameter of the quality estimation function $\Psi$ |
| $\hat{q}$ | A quality estimate made by the estimator $\Psi$ |
| $\cap$ | The verifiable spaces of the estimator's $\Psi$ input parameters $\cap \in \{\{\mathbb{X}\}, \{\mathbb{F}\}, \{\mathbb{X}, \mathbb{F}\}\}$ |

#### A Meta-Evaluation Framework

| | |
|---|---|
| $\mathbb{U}$ | The unverifiable spaces of the estimator's $\Psi$ input parameters $\mathbb{U} \in \{\{\mathbb{E}\}, \{\mathbb{O}\}, \{\mathbb{E}, \mathbb{O}\}\}$ |
| $NR$ | The first failure mode, Noise resilience |
| $AR$ | The second failure mode, Adversary reactivity |
| $y'$ | A prediction after perturbation on the input, model or both input and model spaces |
| $\epsilon$ | A threshold $\epsilon \in \mathbb{R}$ for determining the type of perturbation |
| $t$ | The perturbation strength $t \in \{M, D\}$ |
| $\mathcal{P}_{\cap}(\boldsymbol{\omega})$ | A perturbation function of the verifiable spaces $\boldsymbol{\omega} \in \cap$ |
| $\mathcal{P}_{\mathbb{X}}^{M}$ | A minor perturbation function of the input space $\mathbb{X}$ |
| $\mathcal{P}_{\mathbb{F}}^{M}$ | A minor perturbation function of the function space $\mathbb{F}$ |
| $\mathcal{P}_{\mathbb{X}}^{D}$ | A disruptive perturbation function of the input space $\mathbb{X}$ |

| | |
|---|---|
| $\mathcal{P}_{\mathbb{F}}^D$ | A disruptive perturbation function of the function space $\mathbb{F}$ |
| $K$ | The number of perturbations |
| $L$ | The set of explanation methods |
| $\hat{\boldsymbol{q}}$ | The unperturbed quality estimates $\hat{\boldsymbol{q}} \in \mathbb{R}^N$ |
| $\boldsymbol{q}_k'$ | The perturbed quality estimates, replicated $K$ times for $N$ test samples |
| $d$ | A statistical significance function that takes $\hat{\boldsymbol{q}}$ and $\boldsymbol{q}_k'$ and returns a p-value |
| $r$ | A ranking function that takes nominal values and returns rankings in descending order |
| $\boldsymbol{Q}$ | A matrix of all perturbed samples over $K$ perturbations |
| $\bar{\boldsymbol{Q}}$ | A matrix for the unperturbed estimates $\hat{\boldsymbol{q}}$ for $L$ explanation methods, averaged over $K$ |
| $\bar{\boldsymbol{Q}}'$ | A matrix for the perturbed estimates $\boldsymbol{q}_k'$ for $L$ explanation methods, averaged over $K$ |
| $\bar{\boldsymbol{Q}}^M$ | A matrix for the perturbed estimates under minor perturbation |
| $\bar{\boldsymbol{Q}}^D$ | A matrix for the perturbed estimates under disruptive perturbation |
| $\boldsymbol{U}$ | A binary ranking agreement matrix that takes quality estimates from $\bar{\boldsymbol{Q}}$ and $\bar{\boldsymbol{Q}}'$ and populates the entries according to the interpretation of ranking |
| $\boldsymbol{U}^M$ | A binary ranking agreement matrix with perturbed estimates under minor perturbation |
| $\boldsymbol{U}^D$ | A binary ranking agreement matrix with perturbed estimates under disruptive perturbation |
| $\boldsymbol{m}$ | A meta-consistency vector containing the IAC and IAC scores for both failure modes |
| $\boldsymbol{m}^*$ | An optimally performing quality estimator $\Psi$ as defined by the all-one vector $\mathbb{1}^4$ |
| **IAC** | The intra-consistency scoring criterion, with IAC $\in [0, 1]$ |
| **IEC** | The inter-consistency scoring criterion, with IEC $\in [0, 1]$ |
| **MC** | The meta-consistency score, with MC $\in [0, 1]$ |

### Practical Evaluation

| | |
|---|---|
| $\mathcal{U}$ | The uniform distribution with parameters $\alpha, \beta$ |
| $\alpha$ | The lower bound of the uniform distribution $\mathcal{U}(\alpha, \beta)$ |
| $\beta$ | The upper bound of the uniform distribution $\mathcal{U}(\alpha, \beta)$ |
| $\boldsymbol{\delta}_i$ | Additive uniform noise applied to input space such that $\hat{\boldsymbol{x}}_i = \boldsymbol{x} + \boldsymbol{\delta}_i$ |
| $\mathcal{N}$ | The normal distribution with parameters $\boldsymbol{\mu}, \Sigma$ |
| $\boldsymbol{\mu}$ | The mean of the normal distribution $\mathcal{N}(\boldsymbol{\mu}, \Sigma)$ |
| $\Sigma$ | The variance of the normal distribution $\mathcal{N}(\boldsymbol{\mu}, \Sigma)$ |
| $\boldsymbol{\nu}_i$ | Multiplicative Gaussian noise applied applied to model parameters such that $\hat{\boldsymbol{\theta}}_i = \boldsymbol{\theta} \cdot \boldsymbol{\nu}_i$ |

