# OpenReview forum: "The Meta-Evaluation Problem in Explainable AI: Identifying Reliable Estimators with MetaQuantus"
_TMLR — Accepted by TMLR_

### Review · Reviewer_pFfR · 2023-02-26

**Summary Of Contributions:**

With the increasing use of neural network models in high-stakes applications, XAI methods have been used to describe the behavior of complex models. However, one of the key long-standing problems in XAI is estimating the performance of explanation methods for classification models on a plethora of evaluation metrics. In this work, the authors propose *MetaQuantus* -- a flexible framework that meta-evaluates two failure modes of explanation methods, i.e., noise resilience and adversarial reactivity. Further, the authors show the effectiveness of MetaQuantus by utilizing it to select and optimize hyperparameters of a given evaluation metric.

**Audience:**

Yes

**Broader Impact Concerns:**

The paper is timely and talks about the evaluation of explanation methods which lacks consensus in the XAI community.

**Claims And Evidence:**

Yes

**Requested Changes:**

It would be great if the authors can emphasize the novelty of their work in comparison to existing benchmarking efforts. Please look at the weakness section for more details.

**Strengths And Weaknesses:**

**Strengths**

1. Across multiple explanation methods, datasets, and predictive models, the authors experimentally show that their proposed framework, MetaQuantus, can solve a variety of XAI-related tasks like selecting and optimizing the metrics hyperparameters.

2. The paper provides a robust analysis of different combinations of methods and metrics. In particular, the results of comparing different estimators and performance categories are insightful.

**Weaknesses**

1. The authors can improve the paper writing. In particular, the novelty of the work is unclear as previous benchmarking efforts [1-5] in XAI have already discussed the challenges of the evaluation of XAI methods and the authors list this as one of their contributions.

2. The authors mention that the proposed MetaQuantus has a useful theoretical foundation but it lacks novelty. For instance, the perturbation analysis with respect to input parameters and model parameters has been performed in previous XAI works, showing the sensitivity of explanation methods across different models, explanation methods, and evaluation metrics. Further, the paper does not provide any theoretical analysis of explanation methods for the two proposed failure modes.

3. One of the major drawbacks of MetaQuantus is that it is fundamentally based on perturbation-based metrics (noise resilience and adversarial reactivity) but fails to talk about the advances in the stability analysis of explanation methods. For example, Agarwal et al. [6] refine the original stability metric, proposes three types of relative stability, and argue that the vanilla stability analysis only assumes black-box access to the predictive model, and doesn’t leverage potentially meaningful information such as the model’s internal representations
to evaluate stability.

4. The links to the code do not work.

**References**
1. Hedström et al. Quantus: an explainable ai toolkit for responsible evaluation of neural network explanations. JMLR, 2023
2. Shap benchmark. URL https://shap.readthedocs.io/en/latest/index.html
3. Yang et al. Synthetic benchmarks for scientific research in explainable machine learning. In NeurIPS Datasets and Benchmarks Track, 2021.
4. Agarwal et al. OpenXAI: Towards a Transparent Evaluation of Post hoc Model Explanations. In NeurIPS Datasets and Benchmarks Track, 2022.
5. Faber et al. When comparing to ground truth is wrong: On evaluating GNN explanation methods. KDD, 2021.
6. Agarwal et al. Rethinking Stability for Attribution-based Explanations, ICLR 2022 PAIR^2Struct Workshop, 2022.

---

> ### Author Response · Authors · 2023-03-16
> **General remarks**
>
> We thank reviewer pFfR for all the time and effort taken in reviewing our work!
>
> We are very happy to read that you found our framework and experiments to be robust and according to our claims. We are also glad that you agree with our perspective that the evaluation of explanation methods is a vital concern for the XAI/ ML community, deserving attention and more investigation.
>
> As soon as we have received the third review we will upload the revised manuscript. We will notify all reviewers once the upload is made!

---

> ### Author Response · Authors · 2023-03-16
> **1. Clarify novelty and references**
>
> From our understanding of your review, your first concern is that we did not adequately explain the novel aspects of our solution and neglected to mention other relevant XAI research. We would like to provide some clarification in this regard. While we acknowledge the relevance of the references you suggested ([1], [2], [4] and [6] are works known to us and [1] was already cited) our work differs from them in the following fundamental ways:
>
> References [1], [3] and [4] are different in that they are solving a different problem. These libraries provide metrics, tools or solutions for benchmarking and evaluating XAI methods, whereas meta-evaluation (and MetaQuantus) is a framework for benchmarking the XAI metrics themselves. Reference [2] provides Shapely explanations and differs from our work as they provide explanations rather than running meta-evaluation of XAI metrics. Reference [5] differs from our work in that it focuses on evaluating XAI methods in a scenario where ground truth explanation labels (edges) are generated specifically for GNN models. In contrast, our solution is agnostic to the model type and is moreover, focused on the scenario where the model is considered black-box, implying that the explanations labels remain unknown.
>
> According to your suggested references, we updated the sections “Introduction” and “A Meta-Evaluation Framework” with more comprehensive citations. Additionally, we included an additional sentence in the “Related Works” section to clarify that our work is not an XAI evaluation study, but rather an XAI meta-evaluation contribution. We hope that these changes result in more accurate positioning of our contribution to the XAI community.

---

> ### Author Response · Authors · 2023-03-16
> **2. Theoretical focus**
>
> We thank the reviewer for their feedback on the second requested change. We agree and acknowledge that perturbation analysis with respect to input parameters and/ or model parameters is not novel in the XAI- or XAI Evaluation communities (e.g., see Alvarez-Melis et al., 2018, Yeh et al., 2019, Montavon et al., 2018, Bykov et al., 2021, all which are cited in the paper). The novelty of our work with respect to perturbation analysis is that we (1) combine two different types of perturbation (articulated through failure modes) and (2) apply the ideas of perturbation for the purpose of meta-evaluation of XAI metrics (which to our knowledge, we are the first ones to define in XAI). Through the framing of reliability analysis, we are able to circumvent the lack of ground truth (as stated in the “Challenge of Unverifiability” section) and by using two failure modes and two scoring criteria, we extend previous efforts to analyse XAI metrics (see our “Related Works” section), providing a more comprehensive solution. In order to re-emphasise our novelty, we have revised the paper by adding clarification to the “Abstract” as well as the “Introduction” sections.
>
> Reviewer pFfR further raised the issue of a lack of theoretical analysis of explanation methods for the proposed failure modes. Although we recognise the importance of conducting a deeper theoretical analysis of explanation methods, as demonstrated by recent work such as Binder et al. (2022)'s investigation of the Adebayo model perturbation-based evaluations, we would like to emphasise that our paper focuses on analysing XAI metrics and not the explanation methods. Ultimately, our goal is to establish a more reliable and thorough process for selecting XAI methods, however, before we can accomplish that, we must first address the issue of identifying a reliable XAI metric. Our proposed framework, MetaQuantus, serves as our solution to this problem.

---

> ### Author Response · Authors · 2023-03-16
> **3. Discussion internal representations and drawbacks of perturbation-based analysis**
>
> As the third point, it is valid that the paper does not discuss advances in stability analysis of explanation methods by Agarwal et al. (2022). In the publication, they propose a refinement of the stability/robustness metric and argue for the use of internal representations to evaluate the stability of explanations (and not only with respect to the data or output logits). We acknowledge the significance of this XAI work and have therefore included it as an additional citation in our section “A Meta-Evaluation Framework”—where similar previous works such as Alvarez-Melis et al., 2018, Yeh et al., 2019 and Montavon et al., 2018 have already been referred to—as relevant contributions.
>
> **Discussion of internal representations.** We appreciate the suggestion of reviewer pFfR to incorporate the internal representation of a model (similar to Agarwal et al., 2022). Theoretically, our framework is compatible with the evaluation of latent states or internal representations since internal representations of a certain layer $l$ can be defined as a sub-model $h$, which is part of a compound model $f = h \circ g$. Therefore, by adjusting Definitions 1 and 2 to measure the difference in internal representations between unperturbed $h(\mathbf{x})$ and perturbed sample $h( \mathcal{P}^t_{\mathbb{X}}(\mathbf{x}))$, MetaQuantus and the corresponding software could accommodate internal representations. Incorporating the definition of internal representations to measure the strength of perturbation would, however, require determining a suitable distance metric for comparison and defining boundary conditions for minor versus disruptive perturbations. As discussed by Bykov et al., 2022, answering such questions is not straightforward. Accordingly, we will keep Definitions 1 and 2 unchanged and investigate extension to internal representations in our follow-up work. Our current definitions ensure the general applicability of our framework to any black-box network structure and eliminate the need for access to internal representations while evaluating XAI metrics.
>
> **Drawbacks of perturbation-based techniques.** In response to the reviewer's concern about the potential drawbacks of relying on perturbation-based techniques, we aim to demonstrate in the section titled "Challenge of Unverifiability" that these techniques can still be valuable for studying XAI metrics when ground truth labels are not obtainable. In this section, we further explain (1) why the alternative—validity analysis cannot be performed and (2) how perturbation analysis allows us to investigate the statistical behaviour of XAI metrics from a reliability perspective: by performing multiple runs and studying the intra- and inter-consistency of the XAI metric results, we can determine whether an XAI metric is reliable. If the results are consistent across multiple runs, we can conclude that the metric is reliable. On the other hand, if the results are inconsistent or show significant variation across multiple runs, we can conclude that the metric is not reliable. As we see it, perturbation analysis is a crucial tool that allows us to address the Challenge of Unverifiability and draw meaningful conclusions about the performance of various XAI metrics.
>
> Nonetheless, we are aware that perturbation analysis has its own limitations, particularly in XAI evaluation. For example, Hase et al., 2018 discuss the risk of creating out-of-distribution samples while perturbing the input for XAI evaluation. In our work, we address these limitations, however, by relying on Definitions 1 and 2 to assess the strength of the perturbations within a verifiable space (using the change in predictions as an indicator of perturbation strength).

---

> ### Author Response · Authors · 2023-03-16
> **4. Anonymised links to code**
>
> We apologise for any inconvenience caused by the broken links! The links in the paper were anonymised to uphold author anonymity, as required by the TMLR submission guidelines, and therefore do not work as intended. However, we did include a zipped folder with the carefully anonymised implementations in the Supplements for interested reviewers. Please have a look there.

---

> ### Author Response · Authors · 2023-03-16
> **Final remarks and references**
>
> Thank you so much for all the time taken for this review! If you have any further concerns or suggestions, please don't hesitate to let us know so that we can provide clarification or make any additional changes as needed.
>
> **References**
> * Adebayo et al., (2018) "Sanity Checks for Saliency Maps" https://papers.nips.cc/paper/2018/file/294a8ed24b1ad22ec2e7efea049b8737-Paper.pdf
> * Alvarez-Meliz et al., (2018) "Towards Robust Interpretability with Self-Explaining Neural Networks" https://proceedings.neurips.cc/paper/2018/file/3e9f0fc9b2f89e043bc6233994dfcf76-Paper.pdf
> * Binder et al., (2022) "Shortcomings of Top-Down Randomization-Based Sanity Checks for Evaluations of Deep Neural Network Explanations" https://arxiv.org/abs/2211.12486
> * Bykov et al., (2021) "NoiseGrad — Enhancing Explanations by Introducing Stochasticity to Model Weights" https://ojs.aaai.org/index.php/AAAI/article/view/20561
> * Bykov et al., (2022)  "DORA: Exploring outlier representations in Deep Neural Networks" https://arxiv.org/abs/2206.04530
> * Hase et al., (2021) "The Out-of-Distribution Problem in Explainability and Search Methods for Feature Importance Explanations" https://openreview.net/pdf?id=HCrp4pdk2i
> * Montavon et al., (2018) "Methods for interpreting and understanding deep neural networks" https://www.sciencedirect.com/science/article/pii/S1051200417302385
> * Yeh et al., (2019) "On the (In)fidelity and Sensitivity for Explanations" https://arxiv.org/abs/1901.09392
> * Wickstroem et al., (2021) "RELAX: Representation Learning Explainability" https://arxiv.org/abs/2112.10161

---

### Review · Reviewer_8A54 · 2023-03-03

**Summary Of Contributions:**

The paper presents a framework to evaluate metrics measuring the quality of explainable AI (XAI) methods.
The framework aims to help users to select XAI methods based on given tasks and datasets. Further, it could help to optimize a metric’s hyperparameters.

The present paper claims that the framework is widely applicable across various data-, models-, explanation methods- and metric domains. To this end, the framework measures the influence of "minor" and "disruptive" perturbations. Since different metrics ranking can strongly vary, two consistency criteria are introduced, finally leading to a single Meta-Consistency (MC) score. Empirical evidence is provided by experiments conducted on image classification tasks on the datasets MNIST and fMINST and variations. Multiply XAI evaluation metrics are investigated using the introduced framework. The framework is limited to post-hoc attribution methods.

**Audience:**

Yes

**Broader Impact Concerns:**

There are no concerns present.

**Claims And Evidence:**

No

**Requested Changes:**

1. While the authors mention the limited evaluation in Section 7, other tasks and networks (e.g., transformers and their corresponding novel XAI methods) would strengthen the paper and its impact. Further, it is contrary to what is claimed in the introduction. So I recommend either lowering claims or providing additional evidence.

2. In Sec 2.1:
"[…] we compute the explanation error, requiring a ground truth explanation e. These labels are, however, generally not available for complex ML models and in particular NNs, since their inner workings are considered unknown." Why are ground truth explanations not available for complex ML models? A ground truth explanation would depend on the dataset and task and not on the model. Further, how is complex defined? Is it a two-hidden-layer NN complex or a 175B parameter model complex? Is there a general difference, and how would ground truth explanations using attribution maps differ?

3. Can you further clarify the statement "since most explanation methods and metrics have been developed for the task of image classification"? A large number of methods as well as metrics were also developed for or applied to language processing tasks as well as other vision tasks such as VQA.

**Strengths And Weaknesses:**

**Strength:**

The paper is well written and motivated, including referencing relevant related work and discussing current issues of XAI evaluation metrics.
XAI methods and their evaluation are highly relevant, keeping the impact of current deep learning systems in mind. The framework has the potential to be an important tool in future research.
A wide range of XAI evaluation metrics is empirically investigated, providing novel insights into their advantages and disadvantages.

**Weaknesses:**

Unfortunately, the framework is not evaluated on SOTA deep networks and is limited to image classification tasks (and only convolutional networks), including syntactic datasets. This is, in general, not a major concern. However, especially conflicts with the claims contained in the introduction.
E.g., in this regard, it confuses that the paper claims to evaluate metrics based on the ImageNet dataset. At the same time, this is not contained in the main text’s evaluation and only utilized in a hyperparameter optimization in Appendix Fig. 8. Statements in the introductions such as "wide applicability across various data-, models-, explanation methods- and metric domains" and "a series of experiments on a variety of SOTA explanation methods, datasets and models" are misleading.

---

> ### Author Response · Authors · 2023-03-16
> **General remarks**
>
> We appreciate reviewer 8A54 for their time taken and their great attention to detail shown in the review. We are glad to hear your positive feedback, that you consider our paper well-written and motivated, and that we are addressing current issues in XAI evaluation! We are also pleased to hear that you see potential in our framework as an important tool for future research and found our empirical investigations valuable in this regard.
>
> In the following comments, we respond to the requested changes and focus on the weaknesses mentioned. As soon as we have received the third review we will upload the revised manuscript. We will notify all reviewers once the upload is made!

---

> ### Author Response · Authors · 2023-03-16
> **1. Correct claims and new experiments**
>
> We thank you for pointing out the confusion caused by the inconsistency between the “Introduction” section and the experiments performed in the paper. As a response to your request, we have (1) provided additional evidence regarding the framework’s agnosticism to the choice of network architecture and data by extending the experiments in the paper and (2) revised the claims in the “Introduction” section to improve the reflection of the scope of our experiments. In this process, we also updated the “Experimental Setup” section to clarify, which experimental setups and datasets are used in the main manuscript versus in the Appendix.
>
> **New experiments.** With respect to (1), we conducted additional experiments on ImageNet with transformers-based model architectures. Results are available in the revised version, see Appendix. In particular, we compared XAI metrics of the Localisation and Complexity categories between Vision-Transformer, SWIN and ResNet models. In this regard, reviewer 8A54 also recommended incorporating "novel XAI methods" for transformer-based models, which we considered (e.g., self-attention maps (Caron et al., 2021) and attention backpropagation combined with roll-out (Chefer et al., 2021)). However, it is currently debatable whether attention mechanisms in transformer-based models provide meaningful explanations for a model's decision (Sarthak et al., 2019; Wiegreffe et al., 2019; Pruthi et al., 2020), we therefore use the set of established XAI methods already included in the paper. That being said, the framework (along with the MetaQuantus software) still remains agnostic to the choice of explanation function that outputs $\hat{\mathbf{e}} \in \mathbb{R}^D$ where an investigation of transformers in combination with attention-based techniques is possible as well.
>
> On reviewer 8A54’s suggestion of extending the experiments to additional tasks (such as NLP and VQA)—we agree that such experimentation would be beneficial for the reader to understand the generalisability of the framework. However, as we write in our “Conclusion” section, this expansion does pose additional research questions, especially around the parameterisation of XAI metrics (e.g., for NLP, how to define a suitable input perturbation that satisfies the requirements of the robustness or faithfulness explanation categories) which the XAI evaluation community has not yet fully understood. (We're already working on answering these questions in follow-up work, however, these investigations require further detailed discussions, so we consider them beyond the scope of this paper.)

---

> ### Author Response · Authors · 2023-03-16
> **2. Explanation purpose, existence of ground truth and model complexity**
>
> We thank the reviewer for raising the questions regarding the purpose of explanations, the existence of ground truth explanations and the definition of model complexity. We agree that our formulation was not precise enough. In the following, we discuss the questions and the corresponding adjustments in the manuscript.
>
> **Explanation purpose.** Regarding the provided argument that ground truth explanations depend on the task and data and not the model—this fits into a bigger discussion about whether the purpose of the explanation is to be “true to the model” or “true to the data” (e.g., Janzing et al., 2019 and Sturmfels et al., 2020). As we write in Eq. 2, our definition of an explanation method is defined to explain a model’s decision $\Phi(f(x))$ based on a black-box model function $f$ which necessarily implies that ground truth explanations are unobtainable.
>
> **Existence of ground truth.** We recognise several important works which have built ground truth toy- or expert-annotated XAI datasets, where features are designed to be known a priori (e.g., Arras et al., 2020; Yang et al., 2021; Jun et al., 2022). A key observation in this scenario, however, is that ground truth explanation labels can only be approximated but are not guaranteed to align with the model’s exact decision process or features used. Only with perfect knowledge about how the model deals with the available information (or a carefully manually engineered model), can we claim to have true ground truth explanations with respect to the model. Due to this lack of interpretability, most NNs are often considered black-boxes (e.g., Benitez et. al. 1997, Lipton et al., 2016, Doshi-Velez et. al., 2017, Samek et. al., 2019). We have updated formulations in the “Introduction” and “Preliminaries” sections to reflect this foregoing reasoning.
>
> **Model complexity.** When it comes to reviewer 8A54’s questions about how a “complex” model is defined, we agree that this needs to be formulated more precisely in the paper. Determining a model’s level of complexity is not trivial and can be defined through different lenses. For example, one can define the complexity of a model from an architectural standpoint (how many layers, neurons, activation and connection types or how big is the model in size?) or alternatively, from a perceived complexity from a user standpoint (does this model appear complex or uninterpretable?). As we see it, it is unlikely that setting either an architectural boundary or a user-defined boundary would give a satisfactory answer. As such, we have revised the paper to refrain from using formulations such as “complex models” and instead use “black-box models”, which can generally be understood as "a system of which behaviour is not explicitly known, but which can be observed through its inputs and outputs” (Benitez et al., 1997). This means that the framework applies to models that are characterised by their inscrutability and not their perceived or architectural complexity. If the model is understood from its input-output behaviour rather than its internal workings, it is considered inscrutable.

---

> ### Author Response · Authors · 2023-03-16
> **3. Clarification of statement**
>
> Thank you for bringing this to our attention. We agree that our statement was too narrow and did not fully capture the breadth of XAI methods and metrics that have been developed for various tasks. While we originally made the statement with the understanding that the majority of XAI research has been demonstrated in vision tasks, we acknowledge that many XAI methods and metrics can be applied agnostically across different domains and also, that many XAI methods and metrics were developed e.g., for NLP or VQA specifically. We have removed the statement from the ”Conclusion”.

---

> ### Author Response · Authors · 2023-03-16
> **Final remarks and references**
>
> Once again, we sincerely appreciate the time you've taken to review our manuscript. We hope that we have satisfactorily addressed your concerns and that the changes we've made to the manuscript (which will be shared after the third review) will improve the clarity of our claims and overall content. Please let us know if you have any further concerns, so that we may get the opportunity to provide clarification and/ or make any additional changes!
>
> **References**
> * Arras et al., (2021) "CLEVR-XAI: A benchmark dataset for the ground truth evaluation of neural network explanations" https://www.sciencedirect.com/science/article/pii/S1566253521002335
> * Benitez et al., (1997) "Are artificial neural networks black boxes?" https://pubmed.ncbi.nlm.nih.gov/18255717/
> * Caron et al., (2021) "Emerging Properties in Self-Supervised Vision Transformers" https://openaccess.thecvf.com/content/ICCV2021/papers/Caron_Emerging_Properties_in_Self-Supervised_Vision_Transformers_ICCV_2021_paper.pdf
> * Chefer et al., (2021) "Generic Attention-model Explainability for Interpreting Bi-Modal and Encoder-Decoder Transformers" https://openaccess.thecvf.com/content/ICCV2021/papers/Chefer_Generic_Attention-Model_Explainability_for_Interpreting_Bi-Modal_and_Encoder-Decoder_Transformers_ICCV_2021_paper.pdf
> * Doshi-Velez et al., (2017) "Towards a rigorous science of interpretable machine learning." https://arxiv.org/abs/1702.08608
> * Guidotti et al., (2018) "A survey of methods for explaining black box models." https://dl.acm.org/doi/abs/10.1145/3236009
> * Janzing et al., (2019) "Feature relevance quantification in explainable AI: A causal problem" http://proceedings.mlr.press/v108/janzing20a/janzing20a.pdf
> * Lipton et al., (2016) "The Mythos of Model Interpretability" https://dl.acm.org/doi/10.1145/3236386.3241340
> * Pruthi et al., (2020) "Learning to Deceive with Attention-Based Explanations"https://aclanthology.org/2020.acl-main.432/
> * Samek et al., (2019) "Explainable AI: Interpreting, Explaining and Visualizing Deep Learning" https://link.springer.com/content/pdf/10.1007/978-3-030-28954-6.pdf
> * Sarthak et al., (2019) "Attention is not Explanation" https://aclweb.org/anthology/papers/N/N19/N19-1357/
> * Sturmfels et al., (2020) "Visualizing the Impact of Feature Attribution Baselines." https://distill.pub/2020/attribution-baselines/
> * Wiegreffe et al., (2019) "Attention is not not Explanation" https://aclanthology.org/D19-1002/
> * Yang et al., (2019) "Benchmarking Attribution Methods with Relative Feature Importance" https://arxiv.org/abs/1907.09701

---

### Review · Reviewer_W9T7 · 2023-03-22

**Summary Of Contributions:**

This paper is proposing an evaluation framework for explanation methods, particularly feature based explanation methods. The two main evaluations studied are around: i) how resilient an explanation method is to noise, i.e. small perturbations. ii) should result in small perturbations in explanations and ii) whether an explanation method significantly responds to dsruptive perturbations. Based on this, two "meta"-evaluations have been suggested, on data input and model parameters. Finally the main meta evaluation consists of Intra vs Inter consistency evaluations of these explanation methods to measure these metrics before and after the perturbations.

**Audience:**

Yes

**Broader Impact Concerns:**

I believe the authors sufficiently address part of the broader impact concerns but see above for other comments.

**Claims And Evidence:**

No

**Requested Changes:**

Please see my concerns above regarding the utility and scope of the evaluation framework proposed.
I will reassess my evaluation based on how authors address those concerns.

**Strengths And Weaknesses:**

Strengths:
1. Interesting method to capture explanation quality
2 The motivation seems reasonable and paper is overall well written

Weaknesses:
1. Clearly this evaluation is designed for feature based explanations, although there are many types of explanations. I think the title and content should reflect this fact.
2. Experiments are fairly limited. Not sure the hyper focus on MNIST is as desirable. As a result assessing generalizability of the metrics is a bit challenging.
3. A bigger challenge of explanations is not that there is no unifying way to evaluate them, its that conceptually they are technically estimated in very different ways, making different assumptions, and prone to different types of failure models. Hence the biggest problem for a practitioner is actually assessing whether or not a particular explanation's assumptions are suitable in their problem context. On the other hand the proposed evaluation framework hides this issue even further.
For example, feature importance methods that sample perturbations from marginals, vs conditionals are assessing wildly different sensitivity of the model and their relevance is quite subjective depending on what the user wants to get out of an explanation. By proposing a framework and claiming to be fairly general, I think this will worsen the problem where the actual problems with explanations are now buried under the abstraction of these metrics. Given these concerns, I strongly urge the authors to highlight very clearly, the limitations of the method they are proposing and why it is useful to think of evaluation in this way.

---

> ### Author Response · Authors · 2023-03-29
> **General remarks**
>
> We appreciate the time taken to review our work! We are happy that reviewer W9T7 finds our presented methods interesting and that our paper is well-written!
>
> Before addressing the listed weaknesses one by one we would like to clarify the overall goal of our paper. In consideration of reviewer W9T7’s statements found in the “Summary of Contributions” such as _“This paper is proposing an evaluation framework for explanation methods”_, we would like to emphasise that our paper focuses on the analysis of existing _XAI quality estimators _(i.e., evaluation methods), and _not the explanation methods_ themselves. As part of the paper’s title _“Identifying Reliable Estimators with MetaQuantus”_ and as stated in the Introduction section of the paper, meta-evaluation is defined as _“the process of evaluating the evaluation method”_. Accordingly, we do not propose any additional XAI metric for evaluating explanation methods and do not aim at unifying existing XAI evaluation techniques.
>
> **Clarification of problem.** We would like to draw reviewer W9T7’s attention to the Introduction section (please see Figure 1). Our research has revealed that, when evaluating explanation methods, different quality estimators of the same category (e.g., faithfulness) may assign different rankings to the same explanation method. This disagreement in evaluation outcomes can cause confusion among practitioners, who may not know which explanation method to use when faced with conflicting evaluation results. Despite a few exceptions (Krishna et al., 2022; Wang & Wang, 2021), the XAI community has paid little attention to this existing evaluation issue. This is highly problematic, as choosing an inferior quality estimator may result in the presentation of a sub-optimal explanation to the end user.
>
> **Clarification of contribution.** In our paper, we contribute by (1) raising awareness of this evaluation disagreement issue in XAI (Introduction section), (2) formalising the evaluation problem from ground-up (Preliminaries section) and (3) providing a theoretically-based and comprehensive solution for identifying a reliable XAI metrics (A Meta-Evaluation Framework section). With our open-source software, any XAI or ML practitioners may use our solution to analyse different quality estimators (not XAI methods) in their problem context.
>
> In light of this misinterpretation of our contribution, we have (1) rewritten the Abstract of our submission and (2) also heightened the emphasis on our problem meta-evaluation in the Introduction and Preliminaries sections, for which we found sentences that may have contributed to a potential misreading. We believe that these changes will alleviate any future misinterpretation of our work! Please let us know otherwise.

---

> ### Author Response · Authors · 2023-03-29
> **1. Focus on “feature-based” explanations**
>
> We thank the reviewer for the suggestion* to better reflect the type of explanation method that is included in this work.
>
> We concur with the reviewer's observation that there is currently a broad range of explanation systems available, including but not restricted to prototypes/example-based explanations (e.g., Looveren et al., 2019), counterfactual and anchors (e.g., Ribeiro et al., 2018) (for local explanations), as well as a diverse set of global explainability methods (e.g., Olah et al., 2017; Bykov et al., 2023). These explanation systems all differ in purpose, critical assumptions, and output (shape of features).
>
> **Clarify the scope of explanations.** As demonstrated in Equation 2, we have defined an explanation method as a function that yields $\hat{\mathbf{e}} \in \mathbb{R}^D$. This implies that our solution is currently relevant to all local explanation functions that provide an explanation in this form. This includes the following techniques (but are not limited to):
> * gradient-based explanations (Saliency, Grad-CAM and derivates),
> * back-propagation-based explanations (LRP, Guided Backprop, DeepLIFT),
> * model-agnostic explanations (SHAP, Occlusion),
> * local surrogate explanations (LIME),
> * attention-based explanations (roll-out, self-attention masks, specific to transformer-based architectures),
> * prototypes/example-based explanations.
>
> We have chosen to concentrate on feature-importance methods in our experiments because (i) they have garnered significant attention and popularity within the ML and XAI communities for various data domains (such as vision, text, and tabular data), and are highly applicable to black-box neural network models, and (ii) these methods can accommodate a diverse range of techniques, as listed above. We acknowledge the importance of meta-evaluating quality estimators for other explanation systems as well (e.g., as mentioned above) and believe that this is an important research area that should be explored in a separate paper.  As such, we have revised the Conclusion section by extending it to additional explanation methods as an area for future work as well as revised the content in the Introduction section to clarify that our solution applies to attribution-based (also referred to as feature-importance) explanation methods.
>
> *We kindly note that in their review, reviewer W9T7 referred to "feature based explanations". In our response, we assume that the reviewer was referring to "feature-importance-based explanations", since all explanations are, in one way or another, defined through the use of features.

---

> ### Author Response · Authors · 2023-03-29
> **2. Breath of experiments, dataset focus and generalisability**
>
> We thank the reviewer for these valuable suggestions!
>
> **Shifted focus to ImageNet.** To enhance the generalisability of our results to real-world or SOTA datasets, we have relocated our findings on the MNIST dataset to the Appendix (please see Table 3) and incorporated benchmarking results for ImageNet into the main manuscript (please see Table 1). In the revised version, we have added Figure 5 and expanded Figures 6 and 7 accordingly. We hope that this alleviates the concern that our results have been highly focused on the MNIST (and its variations: fMNIST and cMNIST) datasets.
>
> **Additional experiments.** Regarding the scope of our experiments, we have also performed further experiments with transformers-based model architectures for ImageNet (as suggested by reviewer 8A54). The outcomes of these experiments are now included in the revised version, which might be of interest to reviewer W9T7 (see Appendix A.6.4). Here, we specifically compared quality estimators for the _Localisation_ and _Complexity_ categories across _Vision-Transformer_, _SWIN_, and _ResNet_ models. For detailed background information on this experiment, please see our official comment to reviewer 8A54, titled _"1. Correct claims and new experiments."_.
>
> **Current experimental scope.** Regarding the current scope of our experiments, we have included four datasets (_ImageNet, MNIST, fMNIST, and cMNIST_), three model types (_ResNets, LeNets_ and in the revised version also, _ViT_ and _SWIN_ transformer-based architectures), various combinations of gradient- and model-agnostic attribution-based explanation methods (_Gradient, GradCAM, Saliency, Occlusion, GradientShap, IntegratedGradients, InputXGradient_), evaluated across 12 quality estimators in five evaluation categories (Faithfulness, Robustness, Complexity, Localisation, and Randomisation).  As detailed in both the main manuscript and the Appendix, we conducted experiments that include (i) benchmarking of quality estimators over 5 evaluation categories, (ii) optimisation of hyperparameters of a given estimator, (iii) evaluating the convergence of the different categories of evaluation methods (_Faithfulness, Robustness, Complexity, Localisation, and Randomisation_), and now in the time of rebuttal, we also included (iv) a meta-evaluation comparison across different network architectures (transformers-based versus ResNets). If reviewer W9T7 has any suggestions for areas requiring further investigation, we would be grateful for their guidance.

---

> ### Author Response · Authors · 2023-03-29
> **3. Discussion of failure modes and evaluation**
>
> We thank you for the many interesting points brought up in this question. We acknowledge that our response to the reviewer’s third point may be a bit lengthy, but we believe that the reviewer’s questions raised multiple crucial and intriguing points that necessitated a thorough and extensive discussion.
>
> **Current issues with explanation methods.** We fully acknowledge the challenges that the XAI community encounters in validating the existing explanation methods, as highlighted in our Introduction section, where we referenced several works (Adebayo et al., 2018 and Sixt et al., 2020) that identified “failure modes”: _“Strong assertions of which explanation methods work and not (Adebayo et al., 2018; Sixt et al., 2020), followed by rebuttals (Sundararajan & Taly, 2018; Yona & Greenfeld, 2021; Binder et al., 2022), are ever-present.”_ We have added a citation and rephrased the sentence above by using the term "failure modes" to emphasise that we are referring to the challenges faced by the XAI community in validating explanation methods.
>
> **Cautionary look at established failure modes.** On the point of failure modes, however, we find it important to point out that we should not take the findings regarding the failure modes of different XAI methods at face value. As we referenced in the paper, more recent "rebuttal works" (Sundararajan et al., 2018; Yona et al., 2021, and Binder et al., 2022) have identified actual empirical confounds and theoretical shortcomings in how these failure modes were established. For example, there are problems with the SSIM measure, the randomisation procedures (Binder et al., 2022) and the choice of post-processing of attributions. Overall, the back-and-forth claims and strong assertions of what explanation methods work and not, are highly problematic as they may lead practitioners of XAI to arrive at incorrect conclusions. The complexity of the XAI field, with its numerous choices of methods and evaluation procedures, makes it challenging for practitioners to navigate and identify a suitable explanation method for their explainability context—exactly as reviewer W9T7 points out. In our opinion, enhanced evaluation transparency, greater rigour in evaluation (e.g., through meta-evaluation and associated reliability analysis), and generally, exercising caution in statements would be beneficial.
>
> **“Explain why it is useful to think of evaluation in this way”.** While it is true that explanation methods may be estimated in different ways and prone to different types of failure modes, as we see it, the use of evaluation metrics provides an important common ground to compare and benchmark different methods and assess their suitability—their strengths and weaknesses, in different explainability contexts. From a practitioner's perspective, it is crucial to ask whether an explanation function that fails to capture the behaviour of the underlying model (i.e., lacks faithfulness) and is not resistant to minor input perturbations (i.e., lacks robustness) can be deemed useful. In assessing the suitability of a certain explanation method, evaluation through quality estimators acts complementary to theoretical investigations such as Sundararajan et al., 2017. Consistent with many other works (Liu et al., 2021; Agarwal et al., 2022; Hedström et al., 2023), we believe that adopting more reproducible practices, such as systematic and transparent benchmarking of explanation methods (and evaluation procedures) is useful.
>
> **The need for meta-evaluation.** With that said, it should be acknowledged that evaluation through quality estimation using, e.g., faithfulness, robustness, and complexity is not free of limitations. As reviewer W9T7 has rightly pointed out, evaluation methods relying on perturbations can present challenges (Hase et al., 2022) and also, the quality of an explanation could be interpreted subjectively (i.e., in a biased manner). This leads us directly to our paper’s motivation—our aim is to meta-evaluate the reliability of these quality estimators. Specifically, in awareness of the many challenges that XAI evaluation entails—as we write in the Introduction section: _“we apply this framework to stress test the estimators”_ in this paper, we investigate whether these estimators hold up under different types of perturbations (minor versus disruptive).
>
> **Limitations of our work.** Our work has limitations that we should have highlighted more prominently. The most critical limitation, in our opinion, is the one already mentioned in the Conclusion section, which is that the reliability of a quality estimator does not imply any intrinsic value or ``validity''. However, in light of this discussion, we recognise the importance of cautioning against overreliance or overinterpretation of the results from the meta-evaluation tests, which we have included in the revised version.

---

> ### Author Response · Authors · 2023-03-29
> **Final remarks and references**
>
> We once again thank reviewer W9T7 for their feedback! Please feel free to ask any additional questions or seek further clarifications that may be necessary following our revision.
>
> **References**
>
> * Adebayo et al., (2018) "Sanity Checks for Saliency Maps" https://papers.nips.cc/paper/2018/file/294a8ed24b1ad22ec2e7efea049b8737-Paper.pdf
> * Agarwal et al., (2022) “Rethinking Stability for Attribution-based Explanations” https://openreview.net/forum?id=BfxZAuWOg9
> * Binder et al., (2022) “Shortcomings of Top-Down Randomization-Based Sanity Checks for Evaluations of Deep Neural Network Explanations” https://arxiv.org/abs/2211.12486
> * Bykov et al., (2023) "DORA: Exploring outlier representations in Deep Neural Networks" https://arxiv.org/abs/2206.04530
> * Hedström et al., (2023) “Quantus: An Explainable AI Toolkit for Responsible Evaluation of Neural Network Explanations and Beyond” https://jmlr.org/papers/v24/22-0142.html
> * Hase et al., (2021) "The Out-of-Distribution Problem in Explainability and Search Methods for Feature Importance Explanations" https://openreview.net/pdf?id=HCrp4pdk2i
> * Krishna et al., (2022) “The Disagreement Problem in Explainable Machine Learning: A Practitioner's Perspective” https://arxiv.org/abs/2202.01602
> * Liu et al., (2021) “Synthetic Benchmarks for Scientific Research in Explainable Machine Learning” https://openreview.net/forum?id=R7vr14ffhF9
> * Looveren et al., (2019) “Interpretable counterfactual explanations guided by prototypes.” https://link.springer.com/chapter/10.1007/978-3-030-86520-7_40
> * Olah et al., (2017. “Feature visualization.” https://distill.pub/2017/feature-visualization/
> * Ribeiro et al., (2018) "Anchors: High-Precision Model-Agnostic Explanations" https://homes.cs.washington.edu/~marcotcr/aaai18.pdf
> * Sixt et al., (2020) “When Explanations Lie: Why Many Modified BP Attributions Fail” http://proceedings.mlr.press/v119/sixt20a.html
> * Sundararajan et al., (2018) "A Note about: Local Explanation Methods for Deep Neural Networks lack Sensitivity to Parameter Values" https://arxiv.org/abs/1806.04205
> * Wang & Wang, (2021) "A Unified Study of Machine Learning Explanation Evaluation Metrics." https://arxiv.org/abs/2203.14265
> * Yona et al., (2021) "Revisiting Sanity Checks for Saliency Maps" https://arxiv.org/abs/2110.14297

---

### Decision · Action_Editors · 2023-05-25

**Recommendation:** Accept with minor revision

**Comment:**

This paper first reveals an important issue in Explainable AI (XAI), i.e., the disagreement between different criteria. To address such an issue, the authors developed the meta-evaluation, which is agnostic to the groundtruth and exploiting the perturbation consistency.

The authors did a good job in addressing the concerns raised from the reviewers, especially adding more empirical study on SoTA models with large datasets.

All the reviewers acknowledge that the paper is technical correct and may be of interest to a group of machine learning researchers.

Therefore, I recommend acceptance with minor revision. The authors should take the suggestions from reviewers to make the paper positioning clearly. Meanwhile, I suggest the authors can move the additional experiments on ImageNet with transformers-based model architectures to main text.

**Audience:**


The paper is considering an interesting issue in Explainable AI (XAI). I believe the XAI community will be interested in this paper.

**Claims And Evidence:**

All the reviewers acknowledge that the paper is technical correct.